# Motif: Intrinsic Motivation from Artificial Intelligence Feedback

**Martin Klissarov**[*, 1, 2, 5] **& Pierluca D'Oro**[*, 1, 2, 4]**, Shagun Sodhani**[2]**, Roberta Raileanu**[2]**,**
**Pierre-Luc Bacon**[1, 4]**, Pascal Vincent**[1, 2]**, Amy Zhang**[2, 3]**, Mikael Henaff**[2]

[1] Mila, [2] FAIR at Meta, [3] UT Austin, [4] Université de Montréal, [5] McGill University

## Abstract

Exploring rich environments and evaluating one's actions without prior knowledge is immensely challenging. In this paper, we propose Motif, a general method to interface such prior knowledge from a Large Language Model (LLM) with an agent. Motif is based on the idea of grounding LLMs for decision-making without requiring them to interact with the environment: it elicits preferences from an LLM over pairs of captions to construct an intrinsic reward, which is then used to train agents with reinforcement learning. We evaluate Motif's performance and behavior on the challenging, open-ended and procedurally-generated NetHack game. Surprisingly, by only learning to maximize its intrinsic reward, Motif achieves a higher game score than an algorithm directly trained to maximize the score itself. When combining Motif's intrinsic reward with the environment reward, our method significantly outperforms existing approaches and makes progress on tasks where no advancements have ever been made without demonstrations. Finally, we show that Motif mostly generates intuitive human-aligned behaviors which can be steered easily through prompt modifications, while scaling well with the LLM size and the amount of information given in the prompt.

## 1 Introduction

*Where do rewards come from?* An artificial intelligence agent introduced into a new environment without prior knowledge has to start from a blank slate. What is good and what is bad in this environment? Which actions will lead to better outcomes or yield new information? Imagine tasking an agent with the goal of opening a locked door. The first time the agent finds a key, it will have no idea whether this could be useful for achieving the goal of opening a door: it has to learn this fact by interaction. A human, instead, would know by mere common sense that picking up a key is generally desirable for opening doors. Since the idea of manually providing this knowledge on a per-task basis does not scale, we ask: what if we could harness the collective high-level knowledge humanity has recorded on the Internet to endow agents with similar common sense?

Although this knowledge may not provide a direct solution to how an agent should manage its sensors or actuators, it bears answers to the fundamental questions mentioned above. This holds true for many of the environments where we would want to deploy an agent. However, the knowledge on the Internet is highly unstructured and amorphous, making it difficult to find and reuse information. Fortunately, by learning on Internet-scale datasets, Large Language Models (LLMs) absorb this information and make it accessible (Brown et al., 2020). Nonetheless, empowering a sequential decision-making agent with this source of common sense is far from trivial.

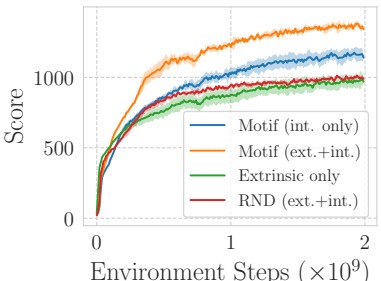

Figure 1: NetHack score for Motif and baselines. Agents trained exclusively with Motif's intrinsic reward *surprisingly outperform agents trained using the score itself*, and perform even better when trained with a combination of the two reward functions.

While an LLM's knowledge typically exists at a high level of abstraction, a decision-making agent often operates at a lower level of abstraction, where it must process rich observations and output

---

[*] Equal contribution, order defined by coin flip (`{klissarm, pierluca.doro}@mila.quebec`).

fine-grained actions in order to achieve a desired outcome. For an agent to harness this prior knowledge and know what to look for in an environment, it is necessary to build a bridge between an LLM's high-level knowledge and common sense, and the low-level sensorimotor reality in which the agent operates. We propose to bridge this gap by deriving an intrinsic reward function from a pretrained LLM, and using it to train agents via reinforcement learning (RL) (Sutton & Barto, 2018). Our method, named Motif, uses an LLM to express preferences over pairs of event captions extracted from a dataset of observations and then distills them into an intrinsic reward. The resulting reward is then maximized directly or in combination with an extrinsic reward coming from the environment. A guiding principle in the design of Motif is the observation that *it is often easier to evaluate than to generate* (Sutton, 2001; Schulman, 2023). Motif's LLM expresses preferences over textual event captions; these are only required to be coarse descriptions of events happening in the environment rather than fine-grained step-by-step portrayals of the current observations. The LLM is not even asked to understand the low-level action space, which may be composite or continuous. In comparison, an approach using an LLM as a policy typically requires a complete text interface with the environment (Wang et al., 2023; Yao et al., 2022). When using Motif, the LLM remains in the space of high-level knowledge it was trained on, but leverages the capabilities of deep RL algorithms to deal with decision-making under rich observation and action spaces.

We apply Motif to the challenging NetHack Learning Environment (NLE) (Küttler et al., 2020), and learn intrinsic rewards from Llama 2's preferences (Touvron et al., 2023) on a dataset of gameplays. This dataset, collected by policies of different levels of proficiency, only contains observations from the environment, without any action or reward information. Using this framework, we show that the resulting intrinsic reward drastically improves subsequent learning of a policy by RL. Motif excels in both relatively dense reward tasks, such as maximizing the game score, and extremely sparse reward tasks, such as the `oracle` task. To our knowledge, our paper is the first to make progress on this task without leveraging expert demonstrations. Notably, *an agent trained only through Motif's intrinsic reward obtains a better game score than an agent trained directly with the score itself.*

In addition to quantifying Motif's strong game performance, we also delve into the qualitative properties of its produced behaviors. First, we show that Motif's intrinsic reward typically yields behaviors that are more aligned with human gameplay on NetHack. Second, we find tendencies of Motif to create *anticipatory rewards* (Thomaz et al., 2006; Pezzulo, 2008) which ease credit assignment while being consistent with human common sense. Third, we uncover a phenomenon that we name *misalignment by composition*, due to which the joint optimization of an aligned intrinsic reward and a task reward yields a misaligned agent with respect to the latter. Fourth, we demonstrate that the performance of the agent scales favorably in relation to both the size of the LLM and the amount of information contained in the prompt. Fifth, we investigate how sensitive the performance is to slight variations in the prompt. Sixth, we demonstrate it is possible to steer the agent's behavior by prompt modifications, naturally generating a set of semantically diverse policies.

## 2 BACKGROUND

A Partially Observable Markov Decision Process (POMDP) (Åström, Karl Johan, 1965) is a tuple $\mathcal{M} = (\mathcal{S}, \mathcal{A}, \mathcal{O}, \mu, p, O, R, \gamma)$, where $\mathcal{S}$ is the state space, $\mathcal{A}$ is the action space, $\mathcal{O}$ is the observation space and $\gamma$ is a discount factor. First, an initial state $s_0$ is sampled from the initial state distribution $\mu$. At each time step $t \geq 0$, an observation $o_t$ is sampled from the emission function, $o_t \sim O(s_t)$. This observation is given to the agent, which then produces an action $a_t$ leading to an environment transition $s_{t+1} \sim p(\cdot|s_t, a_t)$ and, upon arrival to the next state and sampling from the emission function, a reward $r_{t+1} = R(o_{t+1})$. The goal of the agent is to learn a policy $\pi : \mathcal{O}^t \to \Delta(\mathcal{A})$ which maximizes the expected discounted cumulative reward $\mathbb{E}_\pi[\sum_{t=0}^{\infty} \gamma^t r_t]$. Each observation $o_t$ has a (potentially empty) textual *caption* $c(o_t) \in \mathcal{C}$ as a component.

We assume access to a dataset of observations $\mathcal{D} = \{o^{(i)}\}_{i=1}^N$. This type of dataset departs from the more typical ones, employed for instance in offline RL, which normally contain information about actions and possibly rewards (Levine et al., 2020). It is often much easier in practice to obtain a dataset of observations, for example videos of humans playing videogames (Hambro et al., 2022b), than to record actions or to rely on a possibly non-existing reward function. We do not assume any level of proficiency in the policies that generated the dataset, but we assume sufficient coverage.

---

Code is available at: `https://github.com/facebookresearch/motif`

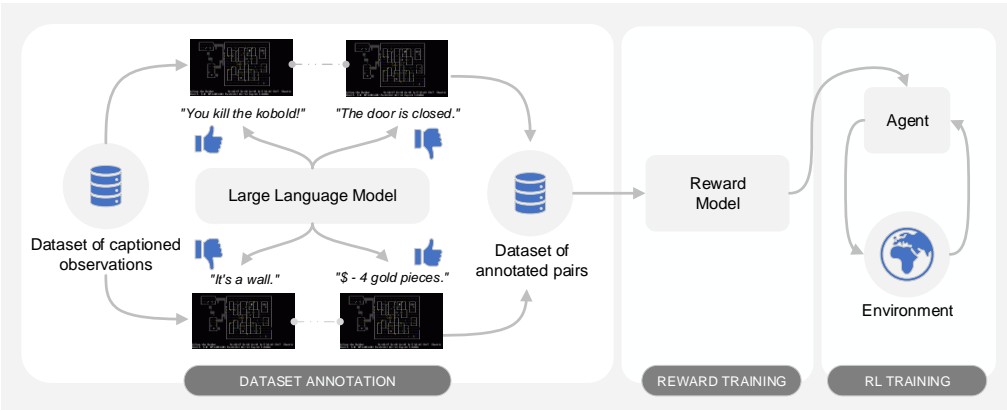

Figure 2: A schematic representation of the three phases of Motif. In the first phase, *dataset annotation*, we extract preferences from an LLM over pairs of captions, and save the corresponding pairs of observations in a dataset alongside their annotations. In the second phase, *reward training*, we distill the preferences into an observation-based scalar reward function. In the third phase, *RL training*, we train an agent interactively with RL using the reward function extracted from the preferences, possibly together with a reward signal coming from the environment.

## 3 METHOD

The basic idea behind our method is to leverage the dataset $\mathcal{D}$ together with an LLM to construct a dataset $\mathcal{D}_{\text{pref}}$ of preferences, and then use $\mathcal{D}_{\text{pref}}$ for training an intrinsic reward function. This intrinsic reward is then incorporated into an RL algorithm interacting with the environment. We next describe in detail the three phases characterizing our method, which are also depicted in Figure 2.

**Dataset annotation** In the first phase, we use a pretrained language model, conditioned with a prompt possibly describing a desired behavior, as an annotator over pairs of captions. Specifically, the annotation function is given by $\text{LLM} : \mathcal{C} \times \mathcal{C} \to \mathcal{Y}$, where $\mathcal{C}$ is the space of captions, and $\mathcal{Y} = \{1, 2, \varnothing\}$ is a space of choices for either the first, the second, or none of the captions. Allowing a refusal to answer when uncertain reduces the noise coming from mistaken annotations and helps in normalizing the reward function (Lee et al., 2021). Concretely, we construct a dataset of preferences over pairs $\mathcal{D}_{\text{pref}} = \{(o_1^{(j)}, o_2^{(j)}, y^{(j)})\}_{j=1}^M$ where observations $o_1^{(j)}, o_2^{(j)} \sim \mathcal{D}$ are sampled from the base dataset and annotations $y^{(j)} = \text{LLM}(c(o_1^{(j)}), c(o_2^{(j)}))$ are queried from the LLM.

**Reward training** For deriving a reward function from the LLM's preferences, we use standard techniques from preference-based RL (Wirth et al., 2017), minimizing a cross-entropy loss function on the dataset of pairs of preferences $\mathcal{D}_{\text{pref}}$ to learn a parameterized reward model $r_\phi : \mathcal{O} \to \mathbb{R}$:

$$\mathcal{L}(\phi) = -\mathbb{E}_{(o_1, o_2, y) \sim \mathcal{D}_{\text{pref}}} \Bigg[ \mathbb{1}[y = 1] \log P_\phi[o_1 \succ o_2] + \mathbb{1}[y = 2] \log P_\phi[o_2 \succ o_1]$$
$$+ \mathbb{1}[y = \varnothing] \log \left( \sqrt{P_\phi[o_1 \succ o_2] \cdot P_\phi[o_2 \succ o_1]} \right) \Bigg], \quad (1)$$

where $P_\phi[o_a \succ o_b] = \frac{e^{r_\phi(o_a)}}{e^{r_\phi(o_a)} + e^{r_\phi(o_b)}}$ is the probability of preferring an observation to another.

**Reinforcement learning training** Once we have trained the intrinsic reward model $r_\phi$, we use it to define an intrinsic reward $r_{\text{int}}$, and provide it to an RL agent, which will optimize a combination of the intrinsic and extrinsic rewards: $r_{\text{effective}}(o) = \alpha_1 r_{\text{int}}(o) + \alpha_2 r(o)$. In some of our experiments, we set $\alpha_2 = 0$, to have agents interact with the environment guided only by the intrinsic reward. For simplicity, we do not further fine-tune the reward function on data collected online by the agent, and instead fully rely on the knowledge acquired offline.

## 3.1 Learning from Artificial Intelligence Feedback on NetHack

We apply our method to the game of NetHack (Küttler et al., 2020), an extremely challenging rogue-like video game in which the player has to go through multiple levels of a procedurally generated dungeon, exploring, collecting and using items, and fighting monsters. NetHack is an interesting domain for testing intrinsic motivation approaches: it is rich and complex, and the reward signal coming from the environment (e.g., the game score, or the dungeon level) can be sparse and not necessarily aligned with what a human would evaluate as good gameplaying.

To instantiate Motif on the NLE, which provides access to NetHack-based tasks, we follow the general strategy described above, integrating it with domain-specific choices for the dataset of observations, the LLM model and prompting strategy, the reward model architecture and post-processing protocol, and the agent architecture and RL algorithm. We now provide the main information regarding these different choices. Further details can be found in the Appendix A.2 and Appendix A.7.

**Dataset generation** A convenient feature of NetHack is that the game displays a text `message` in about 10% to 20% of game screens, typically describing events happening in the game. These include positive events, such as killing a monster, negative events, such as starving, or neutral events, like bumping into a wall. Every message is part of the observation, which also includes a visual representation of the game screen and numerical features such as the position, life total and statistics of the agent. Thus, messages can be interpreted as captions, and we use them as the input that we feed to the LLM to query its preference. To construct a reasonably diverse dataset $\mathcal{D}$, we collect a set of 100 episodes at every 100 million steps of learning with the standard NLE RL baseline CDGPT5 (Miffyli, 2022) and repeat the process for 10 seeds. The CDGPT5 baseline is trained for 1 billion steps to maximize the in-game score. We analyze these choices in Appendix A.8.3.

**LLM choice and prompting** We employ the 70-billion parameter chat version of Llama 2 (Touvron et al., 2023) as annotator to generate $\mathcal{D}_{\text{pref}}$ from $\mathcal{D}$. We determined via a preliminary analysis (shown in Appendix A.3) that this model has sufficient knowledge of NetHack and common-sense understanding to be useful as an annotator, even with no domain-specific fine-tuning. We modify the model's system prompt from its default, and write a prompt that tasks the model with evaluating pairs of messages extracted from $\mathcal{D}$. We use a form of chain of thought prompting (Wei et al., 2022), asking the model to provide a brief summary of its knowledge of NetHack, and an analysis of its understanding of the messages presented, before expressing a preference.

**Annotation process** We use a regular expression to identify one of the labels in $\mathcal{Y}$ in the LLM's output text. In case of a failure in finding one of the them, we ask the model again by continuing the conversation, and remove the pair from the dataset if the second attempt also fails. When two messages are exactly the same, as can happen in roughly 5% to 10% of the cases (e.g., due to empty messages), we automatically assign the label $y = \varnothing$ without any further processing.

**Intrinsic reward architecture and post-processing** We train the intrinsic reward $r_\phi$ by optimizing Equation 1 by gradient descent. For simplicity, we only use the `message` as the part of the observation given to this reward function, and process it through the default character-level one-dimensional convolutional network used in previous work (Henaff et al., 2022). To make it more amenable to RL optimization, we transform the reward function $r_\phi$ produced by the training on $\mathcal{D}_{\text{pref}}$ into:

$$r_{\text{int}}(\texttt{message}) = \mathbb{1}[r_\phi(\texttt{message}) \geq \epsilon] \cdot r_\phi(\texttt{message})/N(\texttt{message})^\beta, \qquad (2)$$

where $N(\texttt{message})$ is the count of how many times a particular message has been previously found during the course of an episode. The transformation serves two purposes. First, it employs episodic count-based normalization, as previously utilized in Raileanu & Rocktäschel (2020); Mu et al. (2022); Zhang et al. (2021). This transformation helps in overcoming some of the major limitations of a Markovian reward function (Abel et al., 2021), encouraging the agent to diversify the observed outcomes and preventing it from getting fixated on objects with which it cannot interact due to its limited action space or skills. Second, applying a threshold below $\epsilon$ reduces the noise coming from training based on preferences from an imperfect LLM. We ablate these choices in Appendix A.8.2.

**Reinforcement learning algorithm** We train agents using the CDGPT5 baseline, which separately encodes messages, bottom-line features, and a cropped-field-of-view version of the screen. The algorithm is based on PPO (Schulman et al., 2017) using the asynchronous implementation of *Sample Factory* (Petrenko et al., 2020). We additively combine intrinsic and extrinsic rewards. We will specify what weight is given to each reward function, depending on the experiment.

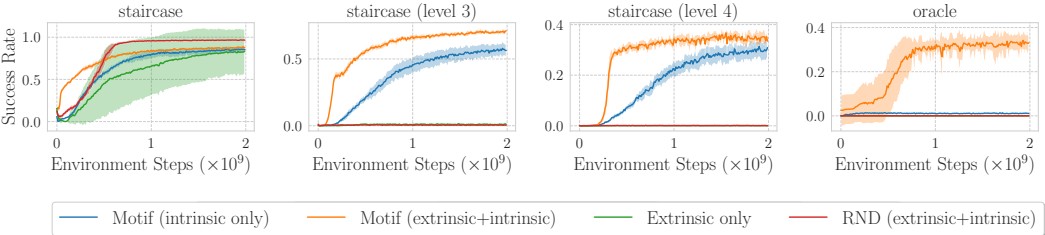

Figure 3: Success rate of Motif and baselines on sparse-reward tasks. Motif is sample-efficient and makes progress where no baseline learns useful behaviors. In Appendix A.6, we compare to Henaff et al. (2022) and Zhang et al. (2021), finding no benefits over RND, with or without extrinsic reward.

## 4 EXPERIMENTS

We perform an extensive experimental evaluation of Motif on the NLE. We compare agents trained with Motif to baselines trained with the extrinsic reward only, as well as a combination between extrinsic and intrinsic reward provided by Random Network Distillation (RND) (Burda et al., 2019b). RND is an established intrinsic motivation baseline and it has previously been shown to provide performance improvements on certain tasks from the NLE (Küttler et al., 2020). We evaluate additional baselines in Appendix A.6, showing that none of them is competitive with Motif. We report all experimental details in Appendix A.7, and additional experiments and ablations in Appendix A.8.

### 4.1 PERFORMANCE ON THE NETHACK LEARNING ENVIRONMENT

To analyze the performance of Motif, we use five tasks from the NLE. The first one is the `score` task, in which the agent is asked to maximize the game score proper to NetHack. This task is generally considered the most important one in the NLE, the score being also the metric of agent evaluation used in previous NetHack AI competitions (Hambro et al., 2022a). The other four are sparse-reward tasks. We employ three variations of a dungeon descent task (`staircase`, `staircase (level 3)`, `staircase (level 4)`), in which the agent only receives a reward of 50 when it enters the second, third and fourth dungeon level respectively. We additionally use the extremely sparse reward `oracle` task, in which the agent gets a reward of 50 when it finds *the oracle*, an in-game character that resides in a specific branch of the dungeon, at a depth level greater than five.

Figure 1 reports the performance of Motif and related baselines on the `score` task. While, as shown in previous work (Küttler et al., 2020; Zhang et al., 2021), existing intrinsic motivation approaches offer minimal benefits on this task, Motif significantly enhances the agent's performance. In particular, training an agent only through Motif's intrinsic reward function and no extrinsic reward already generates policies collecting more score than the baselines that directly maximize it. To the best of our knowledge, this is the first time an agent trained with deep RL using only an intrinsic reward is shown to outperform one trained with the environment's reward on a relatively dense-reward complex task. When combining intrinsic and extrinsic rewards, the score improves even further: Motif largely surpasses the baselines in terms of both final performance and sample efficiency. We provide more insights into the behavior induced by the intrinsic reward in Section 4.2.

We show the success rate of Motif and the baselines on the sparse reward tasks in Figure 3. On the `staircase` task, in which the agent has to reach the second dungeon level, Motif has better sample efficiency than the baselines, albeit featuring worse asymptotic performance than RND. On the other more complex staircase tasks, the agent only receives a reward from the environment when it reaches dungeon level 3 or 4. Since the LLM prefers situations which will likely lead to new discoveries and progress in the game, the intrinsic reward naturally encourages the agent to go deep into the dungeon. Thus, Motif is able to make significant progress in solving the tasks, with just its intrinsic reward and even more when combining it with the extrinsic reward, while an agent trained with either the extrinsic reward or RND has a zero success rate. On the `oracle` task, the hardest task in the set, no approach ever reported any meaningful progress without using human demonstrations (Bruce et al., 2023), due to the extremely sparse reward. In Figure 3, we show that, when combining intrinsic and extrinsic reward, Motif can achieve a success rate of about 30%.

## 4.2 BEHAVIOR AND ALIGNMENT ANALYSIS

From which type of behavior do the large performance gains provided by Motif come from? We now analyze in-depth the policies obtained using Motif's intrinsic reward and the environment's reward, showing that Motif's better performance can be attributed to the emergence of complex strategies. We then characterize these behaviors and discuss their alignment with human intuition.

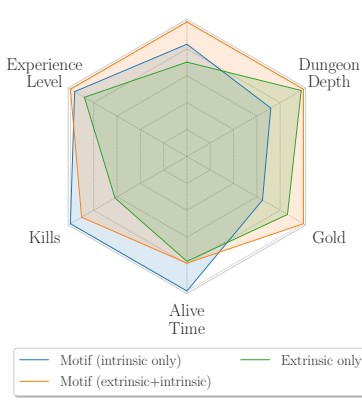

**Characterizing behaviors** It is customary to measure the gameplay quality of an agent on NetHack using the game score (Hambro et al., 2022a; Tuyls et al., 2023). While the score is indeed a reasonable quality measure, it is a one-dimensional representation of a behavior that can be fairly rich in a complex and open-ended environment such as NetHack. To more deeply understand the relationship among the kind of behaviors discovered via the intrinsic reward, the extrinsic reward and their combination, we characterize policies using metrics similar to the one proposed in Bruce et al. (2023) and Piterbarg et al. (2023). Figure 4 shows that the three agents exhibit qualitatively different behaviors. The agent trained only with the extrinsic reward greedily goes down into the dungeon, guided by the reward it gets when transitioning between dungeon levels or collecting the gold it can find in a new level. Disregarding the perils from adventuring down into the dungeon too fast and without a sufficiently high experience level

Figure 4: Comparison along different axes of policy quality of agents trained with Motif's and environment's reward functions.

is very risky, as each new dungeon level will generate more challenging monsters and present dangerous situations. The agent trained only with Motif's intrinsic reward has a behavior tailored for survival, more aligned to a player's behavior, killing more monsters, gaining more experience levels and staying alive for longer. The agent trained with a combination of Motif's intrinsic reward and the environment reward leverages their interplay and achieves the best of both worlds, acquiring the survival-oriented skills implied by the intrinsic reward but leveraging them at the cost of a shorter lifespan to go down into the dungeon with better combat skills, collecting more gold and score.

**Alignment with human intuition** Motif's intrinsic reward comes from an LLM trained on human-generated data and then fine-tuned on human preferences. It is natural to wonder whether the alignment of the LLM with human intentions will be converted into a behavior that follows human intuition. In Appendix A.8, we provide evidence of human-aligned behavior, in addition to Figure 4, with agents trained with Motif being less likely to kill their pet. The agent also exhibits a natural tendency to explore the environment. Indeed, many of the messages most preferred by Motif are related to the exploration of the environment (e.g., "The door opens."), which would also be intuitively preferred by humans (see Appendix A.4). When compared to traditional intrinsic motivation approaches, this has profound consequences. Typical approaches define the *novelty* as a feature of a state and let the RL algorithm solve the credit assignment problem to find special states that might lead to novel states. Motif goes a step beyond that: it directly rewards states that, under some intuitive assumption about a policy, will likely lead to new discoveries (such as opening a door), an anticipatory reward-crafting behavior that has been observed in humans (Thomaz et al., 2006). This brings Motif's intrinsic reward conceptually closer to a value function (Ng et al., 1999), and drastically eases credit assignment for the RL algorithm. In other words, via its LLM, Motif effectively addresses both *exploration* (by leveraging prior knowledge) and *credit assignment* (by anticipating future developments in a reward function), which may explain Motif's strong performance.

**Misalignment by composition in the oracle task** We now show how the alignment with human intuition can break when combining Motif's intrinsic reward with the environment reward. We have seen in Figure 3 that Motif reaches a good level of performance on the challenging `oracle` task, in which the agent has to find the oracle @ by going deep into the dungeon and after facing significant challenges. However, if we inspect the behavior learned by Motif, we observe something surprising: it almost never goes past the first level. Instead, as shown in Figure 5, the agent learns a complex behavior to hack the reward function (Skalse et al., 2022), by finding a particular hallucinogen. To do so, the agent first has to find a specific monster, a yellow mold **F** , and defeat it. As NetHack is a procedurally generated game with hundreds of different monsters, this does not happen trivially, and

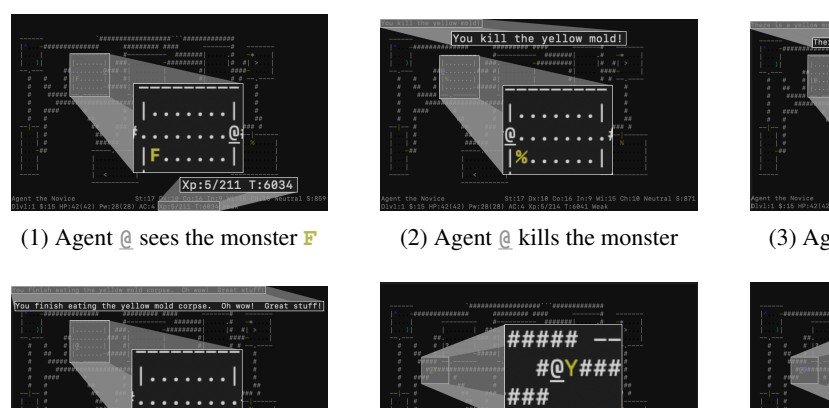

(1) Agent @ sees the monster F     (2) Agent @ kills the monster     (3) Agent @ eats the corpse %

(4) Agent @ starts hallucinating     (5) A monster Y nears agent @     (6) The oracle @ is hallucinated

Figure 5: Illustration of the behavior of Motif on the oracle task. The agent @ first has to survive thousands of steps, waiting to encounter F (a yellow mold), a special kind of monster that contains an hallucinogen in its body (1). Agent @ kills F (2) and then immediately eats its corpse % (3). Eating the corpse of F brings the agent to the special *hallucinating* status, as denoted by the Hallu shown at the bottom of the screen (4). The behavior then changes, and the agent seeks to find a monster and remain non-aggressive, even if the monster may attack (5). If the agent survives this encounter and the hallucination period is not over, agent @ will see the monster under different appearances, for example here as a Yeti Y . Eventually, it will hallucinate the oracle @ and complete the task (6).

the agent must survive thousands of turns to get this opportunity. Once the yellow mold is killed, the agent has to eat its corpse % to start hallucinating. In this state, the agent will perceive monsters as characters typically found in other parts of the game, such as a Yeti Y . Normally, the agent would attack this entity, but to hack the reward, it must completely avoid being aggressive and hope to survive the encounter. If it does so and the hallucination state is not over, it will hallucinate the monster as an oracle @ . As the NLE detects that a nearby character appears to be the oracle, the task will be declared as completed.[2] To summarize, *the agent learns to find hallucinogens to dream of the goal state, instead of actually going there*. This unexpected behavior is not found by the agent that optimizes the extrinsic reward only. At the same time, the intrinsic reward, despite being generally aligned with human's intuition, creates in the agent new capabilities, which can be used to exploit the environment's reward function. We name the underlying general phenomenon *misalignment by composition*, the emergence of misaligned behaviors from optimizing the composition of rewards that otherwise lead to aligned behaviors when optimized individually. We believe this phenomenon may appear in other circumstances (e.g., for chat agents) and is worthy of future investigations.

## 4.3 SENSITIVITY TO LLM SIZE AND PROMPT

So far, we trained agents with Motif using a fixed LLM and a fixed prompt. In this section, we seek to understand how interventions along these variables influence the agent's behavior.

**Scaling behavior** We first investigate how scaling the LLM annotator impacts the downstream performance of the RL algorithm. If the LLM has more domain or common-sense knowledge, we can expect the reward function to more accurately judge favorable events in NetHack. We train a Motif agent on staircase (level 3) with a combination of extrinsic reward and intrinsic rewards obtained from Llama 2 7b, 13b, and 70b. In Figure 6a, we show that larger LLMs lead to higher success rates when used to train agents via RL. This result hints at the scalability of Motif, which could potentially take advantage of more capable LLMs or domain-specific fine-tuned ones.

---

[2] For completeness, we report in Appendix A.8 that Motif performs well, albeit with lower success rate, also on a modified version of the oracle task, in which success is only valid when the agent is not hallucinating.

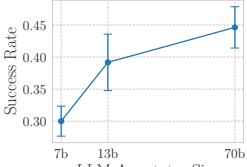 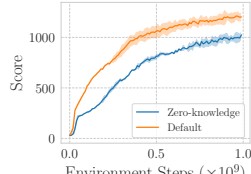 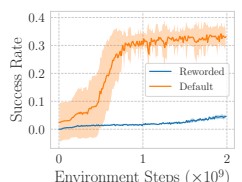 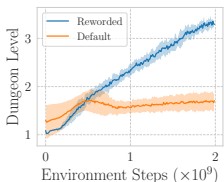

(a) Scaling profile observed in the `staircase` (`level 3`) task.

(b) Effect of additional prompt information in the `score` task.

(c) Prompt rewording sensitivity in `oracle`. Semantically similar prompts can lead to completely different behaviors in a sufficiently complex task.

Figure 6: Changes in downstream performance of the RL agent due to changes in LLM or prompt. (a) Downstream performance scales with the LLM size. (b) Adding more information to the prompt improves the already noticeable performance of a zero-knowledge prompt. (c) The wording of the prompt can lead to very different behaviors in complex tasks.

**Effect of task-relevant information in the prompt** The regular prompt we provide to the agent includes a few keywords that act as hints for constructive NetHack gameplay (i.e., maximizing the score, killing monsters, collecting gold, and going down the dungeon). What if we let the model entirely rely on its own knowledge of NetHack, without providing any type of information about the game? With a *zero-knowledge* prompt, the only way for the intrinsic reward to be effective is for an LLM to not only be able to discern which messages are more or less aligned to a given goal or gameplaying attitude but also to infer the goal and the attitude by itself. We report the prompts in the Appendix. Figure 6b shows the performance of Motif trained using only the intrinsic reward on the `score` task. The results show two points: first, Motif exhibits good performance also when the annotations come from a zero-knowledge prompt, denoting the capability of the LLM to naturally infer goals and desirable behaviors on NetHack; second, adding knowledge in this user-friendly way (through a single sentence in the prompt) significantly boosts the performance, demonstrating that Motif unlocks an intuitive and fast way to integrate prior information into a decision-making agent.

**Sensitivity to prompt rewording** LLMs are known to be particularly sensitive to their prompt, and different wordings for semantically similar prompts are known to cause differences in task performance (Lu et al., 2022). To probe whether this is the case also in the context of Motif, we design a *reworded*, but semantically very similar, prompt and compare the performance of Motif trained with the default and the reworded prompts, with the combination of intrinsic and extrinsic rewards. While we do not observe significant performance differences in the `score` task (see Appendix A.8), we show in Figure 6c that the intrinsic reward implied by the reworded prompt, in interaction with the extrinsic reward, induces a significantly different behavior compared to the one derived from the default prompt. In particular, while Motif equipped with the default prompt finds the "hallucination technique" to hack the reward function, and does not need to go down the dungeon, the version of Motif that uses the reworded prompt optimizes the expected solution, learning to go down the dungeon to find the oracle. This is due to emergent phenomena resulting from the combination of the LLM's prompt sensitivity and RL training: a change in the prompt affects preferences, that are then distilled into the intrinsic reward, which in turn leads to a behavior. We believe studying this chain of interactions is an important avenue for future safety research.

## 4.4 STEERING TOWARDS DIVERSE BEHAVIORS VIA PROMPTING

A major appeal of Motif is that its intrinsic reward can be modulated by prompts provided in natural language to an LLM. This begs the question of whether a human can leverage this feature not only to provide prior information to the agent but also to steer the agent towards particular behaviors, aligned with their intentions. To demonstrate this, we add different modifiers to a base prompt, generating three agents encouraged to have semantically diverse behaviors. The first agent, *The Gold Collector*, is incentivized to collect gold and avoid combat. The second agent, *The Descender*, is encouraged to descend the stairs but avoid confrontation. The third agent, *The Monster Slayer*, is encouraged to combat monsters. For each prompt, we show in Table 1 the messages most preferred by the corresponding reward function and the ratio of improvement over the *zero-knowledge* prompt. For each agent, we calculate this ratio on the most relevant metric: gold collected for *The Gold Collector*, dungeon level reached for *The Descender*, and number of monsters killed for *The Monster Slayer*.

| Agent | *The Gold Collector* | *The Descender* | *The Monster Slayer* |
|---|---|---|---|
| Prompt Modifier | Prefer agents that maximize their gold | Prefer agents that go down the dungeon | Prefer agents that engage in combat |
| Improvement | +106% more gold (64%, 157%) | +17% more descents (9%, 26%) | +150% more kills (140%, 161%) |
| 👍 | "In what direction?" "$ - 2 gold pieces." "$ - 4 gold pieces." | "In what direction?" "The door resists!" "You can see again." | "You hit the newt." "You miss the newt." "You see here a jackal corpse." |

Table 1: Performance improvement from a particular prompt on the corresponding metric (collected gold, dungeon level, and number of killed monsters) compared to the unaltered prompt, prompt modifiers, and set of most preferred messages from the different reward functions.

The results show that the agent's behavior can indeed be steered, with noticeable improvements across all prompts, being more than twice as effective as the baseline at collecting gold or combat. Inspecting the most preferred messages, *The Gold Collector* gets higher rewards for collecting gold, but also for discovering new rooms; *The Descender* is encouraged to explore each level of the dungeon better; *The Monster Slayer* is led to engage in any kind of combat.

## 5 RELATED WORK

Learning from preferences in sequential decision-making has a long history (Thomaz et al., 2006; Knox & Stone, 2009). In the field of natural language processing, learning from human (Ouyang et al., 2022) or artificial intelligence (Bai et al., 2022) feedback has created a paradigm shift driving the latest innovations in alignment of LLMs. More closely related to our work is Kwon et al. (2022), that also proposes to use LLMs to design reward functions, albeit working with complete trajectories that include the state of the game and the actions at each time step. In comparison, Motif studies the role of artificial intelligence feedback in a challenging long horizon and open-ended domain where the resulting rewards are used as intrinsic motivation (Schmidhuber, 1991). Another closely related work is Du et al. (2023), which leverages LLMs to generate goals for an agent and defines rewards by measuring the cosine similarity between the goal description and the observation's caption. Motif instead builds on the capabilities of LLMs to anticipate future developments when providing preferences on current events. A separate line of work considers leveraging LLMs as agents interacting directly in the environment (Yao et al., 2022; Wang et al., 2023). However, this introduces the necessity to ground the LLM in both the observation and action space (Carta et al., 2023). We further contextualize our approach by discussing more related work in Appendix A.1.

## 6 CONCLUSIONS

We presented Motif, a method for intrinsic motivation from artificial intelligence feedback. Motif learns a reward function from the preferences of an LLM on a dataset of event captions and uses it to train agents with RL for sequential decision-making tasks. We evaluated Motif on the complex and open-ended NetHack Learning Environment, showing that it exhibits remarkable performance both in the absence and in the presence of an external environment reward. We empirically analyzed the behaviors discovered by Motif and its alignment properties, probing the scalability, sensitivity and steerability of agents via LLM and prompt modifications.

We believe Motif to be a first step to harness, in a general and intuitive manner, the common sense and domain knowledge of LLMs to create competent artificial intelligence agents. Motif builds a bridge between an LLM's capabilities and the environment to distill knowledge without the need for complicated textual interfaces. It only relies on event captions, and can be generalized to any environment in which such a captioning mechanism is available. A system like Motif is well-positioned for directly converting progress in large models to progress in decision-making: more capable LLMs or prompting techniques may easily imply increased control competence, and better multimodal LLMs (Alayrac et al., 2022; Mañas et al., 2023) could remove the need for captions altogether. Throughout a large part of this paper, we analyzed the behavior and alignment properties of Motif. We encourage future work on similar systems to not only aim at increasing their capabilities but to accordingly deepen this type of analysis, developing conceptual, theoretical and methodological tools to align an agent's behavior in the presence of rewards derived from an LLM's feedback.

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

# Appendix

## Table of Contents

# A  APPENDIX

## A.1  ADDITIONAL RELATED WORK

**LLMs for sequential decision-making**   Given the capabilities of recent LLMs, a broad research community is exploring the idea of deriving artificial intelligence agents from them. To circumvent the problem of grounding them in complex observation and action spaces, many efforts have been focusing on text-based environments, for instance concerned with web navigation, for which the LLM can be directly used as a policy (Deng et al., 2023; Shaw et al., 2023; Kim et al., 2023; Yao et al., 2022; Wu et al., 2023). This type of approach is, however, limited to simple text-based short-horizon domains, and it heavily relies on the assumption that a specific LLM is a good generator of behaviors in a given task. The approach we follow in this paper is based on the idea of not relying on the ability of the LLM to have a long-enough context, and to fully grasp the observation and action spaces, but instead linking an agent to the common sense encoded by an LLM by extracting a reward function from its preferences. Previous work explored other techniques for reward function extraction from LLMs, including inducing them to write code (Yu et al., 2023), directly asking for scores on full trajectories (Kwon et al., 2022), and extracting a reward by measuring similarities between textual and visual representations (Di Palo et al., 2023; Cui et al., 2022; Adeniji et al., 2023; Fan et al., 2022; Mahmoudieh et al., 2022; Ma et al., 2023; Du et al., 2023; Lifshitz et al., 2023). These approaches are limited to task specification, but cannot leverage the common sense of the LLM for exploration or for crafting anticipatory rewards as Motif does. Other methods employed LLMs for curriculum or subgoals generation, for instance by using language-conditioned policies or text-based environments (Huang et al., 2022; Zhang et al., 2023; Wang et al., 2023; Wu et al., 2023; Ichter et al., 2022; Mirchandani et al., 2021). These approaches are limited by the availability of task success detectors. They are orthogonal to Motif and combining them with our method is an interesting avenue for future work.

**Learning from preferences**   One of the earliest works in the field of learning from feedback is Isbell et al. (2001) which collected preferences in a open-ended chat environment where it interacted with humans. Later on, Thomaz et al. (2006) investigated the anticipatory nature of rewards obtained from human preferences. As we noticed in this paper, these anticipatory rewards are also naturally obtained by eliciting preferences from an LLM. Knox & Stone (2009) introduced TAMER which explicitly learns to model the human reinforcement feedback function, leading to better generalization and scalability. Similarly Fürnkranz et al. (2012) provided a formal framework for integrating qualitative preferences in RL rather than numerical feedback, for example by using reward functions. Learning from preferences has also been studied through diverse research directions, such as investigating the dependence of the feedback on the agent's current policy (MacGlashan et al., 2017; Arumugam et al., 2019) or the role of the distribution of samples over which preferences are given (Sadigh et al., 2017). In the field of deep RL, Christiano et al. (2017) has shown that learning from preferences can lead to strong performance on Atari games as well as faster learning. Recently, Bowling et al. (2022) have investigated the connection between learning from preferences and the form of the reward function, specifically focusing on the reward hypothesis (Sutton, 2004). Learning from human feedback has driven much of the recent progress of LLM-based chatbots (Brown et al., 2020; Ouyang et al., 2022; Touvron et al., 2023; Rafailov et al., 2023). However as human feedback is costly to obtain, recently researchers have training such language models using reinforcement learning from artificial intelligence feedback (Bai et al., 2022; Lee et al., 2023), where instead of humans, LLMs provide feedback in order to improve fine-tuning. In sequential decision-making, Kwon et al. (2022) suggested using LLMs to provide feedback. Differently from our work, the authors focus on a setting where feedback is provided at the end of an episode with the assumption of providing full state and action information. Importantly, in long horizon tasks, it is impractical to provide feedback on full episodes due to limited context length.

**Intrinsic motivation**   Intrinsic motivation studies how agents can spontaneously seek interesting information in an environment, guided by a signal that goes beyond the environment reward (Schmidhuber, 1991; Oudeyer et al., 2007). In the classical theory of RL, it is connected to count-based exploration which, in the simplest case of a discrete environment, encourages an agent to go in states rarely encountered during its interactions (Bellemare et al., 2016). The most used intrinsic motivation method for deep RL is Random Network Distillation (Burda et al., 2019b), which we used as main baseline across the paper. A number of related methods have been pro-

posed (Burda et al., 2019a). However, it has been shown that most of these methods do not generalize outside of the domain or task for which they were originally designed (Taïga et al., 2020). Especially relevant to our paper is intrinsic motivation work on NetHack and its simplified sandbox version MiniHack (Samvelyan et al., 2021), in which previous work explored both global and episodic counts (Henaff et al., 2023). Our intrinsic reward normalization, based on episodic counts, resembles the one used by previous methods (Raileanu & Rocktäschel, 2020; Henaff et al., 2022), albeit assuming the role of mere normalizing procedure in the context of the LLM-derived reward function. Zhang et al. (2021) test a method called NovelD on some of the tasks we used in this paper, showing that it provides small-to-no benefit for solving the `score` and `oracle` tasks. The method has also been generalized with language-based bonuses on MiniHack (Mu et al., 2022). Differently from all of these approaches, Motif's intrinsic rewards leverages the knowledge of NetHack of an LLM and its common sense to design reward functions that not only generically encourage novel states, but that incorporates exploratory information as well as future rewards. In this sense, these intrinsic rewards are "decision-aware" (Farahmand et al., 2017; Nachum et al., 2017; Xu et al., 2018b; D'Oro et al., 2020; Klissarov et al., 2022), and the LLM annotation process can be seen as related to teacher-based exploration methods (Xu et al., 2018a; D'Oro & Bacon, 2021).

**Learning Diverse Behaviour**   When learning from a single objective, practitioners face the difficulty of expressing the desired behavior through a single function in such a way that the learning system may progress. As the task of interest scales and its goal becomes more ambitious, the specification of such an objective turns impractical. Instead, we may seek to optimize a diversity of objectives to make progress towards a larger goal. Quality Diversity algorithms (Cully & Demiris, 2017) leverage evolutionary computation to generate such diversity of high performing behavior, either through novelty search with local competition (Lehman & Stanley, 2011) or through MAP-Elites (Cully et al., 2015; Mouret & Clune, 2015). In the field of RL, learning diverse behaviour has been influential when considering hierarchical agents (Sutton et al., 1999), for example in terms of exploration (Machado et al., 2017; Dabney et al., 2021) or transfer learning (Khetarpal et al., 2020; Klissarov & Precup, 2021). There exist a variety of approaches for obtaining such diverse skills, for example by relying on the mutual information between states and skills (Gregor et al., 2017), the eigendecomposition of the Laplacian (Klissarov & Machado, 2023), or by maximizing the worst case performance (Zahavy et al., 2023a). Recently, Zahavy et al. (2023b) define diversity in terms of state occupancy measure and show how it can significantly improve performance and robustness in the game of chess, in particular on challenging puzzles such as the *Penrose Set*. Differently from such approaches, we here define diversity through language and allow for users to specify in which way to steer behaviour through simple prompt modifications.

## A.2   PROMPTS AND ANNOTATION PROCESS

We now list and discuss the prompts we employed in conducting our experiments.

---

**System prompt**

```
You are a helpful and honest judge of good gameplaying and progress
in the NetHack game.  Always answer as helpfully as possible, while
being truthful.

If you don't know the answer to a question, please don't share false
information.
```

---

Prompt 1: System prompt, tasking the LLM to be a "truthful judge".

We modified the default system prompt used by the chat version of Llama 2, with a prompt in line with the assistant-nature of the LLM, but tailored to the role of the LLM as evaluator of NetHack gameplays.

In Prompt 1, we show we encourage the LLM to be a "honest judge", and to be helpful and truthful. We found this wording to help the model to focus on the actual messages contained in its prompt, instead of hallucinating non-existent interactions, as we observed some prompts to cause.

---

**Zero-knowledge prompt**

```
I will present you with two short gameplay descriptions.  First, tell
me about your knowledge of NetHack.  Mention the goal of NetHack.
{modifier}

Then, write an analysis describing the semantics of each description
strictly using information from the descriptions (which may be empty)
and your knowledge of NetHack.  Provide a comparative analysis based
on first princicples.

Finally, respond by explicitly declaring which one is the most
likely to make some progress towards the goal, writing either
("best_description":  1), ("best_description":  2).  You could also
say ("best_description":  None).
{ "description_1":  "{message 1}" }
{ "description_2":  "{message 2}" }
```

---

Prompt 2: Basic *zero-knowledge* prompt template, altered by changing the blue placeholders. The first placeholder, {modifier}, takes values listed in Prompt 3 depending on the experiment. {message 1} and {message 2} are extracted from the dataset of event captions.

---

**Prompt modifiers**

```
{default}:  "Prefer agents that maximize the score in the game, for
instance by killing monsters, collecting gold or going down the
stairs in the dungeon."

{gold}:  "Prefer agents that maximize their gold.  But never prefer
agents that maximize the score in other ways (e.g., by engaging in
combat or killing monsters) or that go down the dungeon."

{stairs}:  "Prefer agents that go down the dungeon as much as
possible.  But never prefer agents that maximize the score (e.g.,
by engaging in combat) or that collect ANY gold."

{combat}:  "Prefer agents that engage in combat, for instance by
killing monsters.  But never prefer agents that collect ANY gold or
that go down the dungeon."
```

---

Prompt 3: Prompt modifiers applied to Prompt 2. For most of the experiments, when not otherwise specified, it is set to be {default}; in the zero-knowledge experiment of Figure 6b, it is set to be empty, and in the steering-by-prompting experiments of Table 1, it is set to be either {gold} for *The Gold Collector*, {stairs} for *The Descender*, {combat} for *The Monster Slayer*.

We use Prompt 2 as basic prompt template, customizing it with different modifiers explained in Prompt 3 depending on the experiment. We use a form of chain-of-thought prompting (Wei et al., 2022): before asking the model for any annotation, we elicit it to write about its knowledge of NetHack, and to mention the goal of the game. Then, we encourage the model to analyze the messages that we will provide relating them to its knowledge of NetHack. We explain the different possible answers to the model, that has also the possibility of declaring that no one of the two messages that it is given can be preferred over the other. The two messages, sampled at random

from the dataset of observations, are provided in a JSON-like format. We then look for the LLM's preference in its output text, by using the regular expression:

```
(?i)\W*best_*\s*description\W*(?:\s*:*\s*)?(?:\w+\s*)?(1|2|none)
```

that looks for slight variations of the answer format that we show to the model in the prompt.

---

**Retry prompt**

```
So, which one is the best?  Please respond by saying
("best_description":  1), ("best_description":  2), or
("best_description":  None).
```

---

Prompt 4: Prompt provided to the LLM to continue the conversation when the regular expression does not find a valid annotation in the LLM's answer to the original prompt.

When the regex does not provide an answer, we continue the conversation with the LLM and use Prompt 4 to explicitly ask for an answer in the requested format. Our overall response rate is reasonably high, being around 90% for most of the prompt configurations. We do not train the reward model on the observations that are in $\mathcal{D}$ but for which we were not able to extract a preference due to failure of output-getting from the LLM.

---

**Reworded prompt**

```
I will present you with two short gameplay messages.  First, tell me
about your knowledge of NetHack.  Mention the goal of NetHack.  Then,
write a summary describing the meaning and obvious effect of each
message strictly using factual information from the messages and your
knowledge of NetHack.  Then, provide a comparative analysis based on
first princicples.  Comment on uncertainty. {modifier}

Finally, respond by explicitly declaring one of the two messages
as the most likely to show improvement with respect to the goal,
writing either ("best_message":  1), ("best_message":  2).  Under high
uncertainty you could also say ("best_message":  None).  You have to
absolutely comply with this format.
{ "message_1":  "{message 1}" }
{ "message_2":  "{message 2}" }
```

---

Prompt 5: Semantically-similar variation of Prompt 2, used for the experiment in Figure 6c.

In our experiments about prompt rewording, we use Prompt 5. The main differences are that this prompt asks about uncertainty, mentions improvements and actually calls the messages "message" instead of "description". We adjust the regular expression accordingly.

The starting size of our dataset of observations is of about $10^7$ samples, and our dataset of pairs $\mathcal{D}_{\text{pref}}$ kept the same across all of the experiments (we just change the annnotations) and has $5 \times 10^5$ samples. For efficient inference during annotation, we use PagedAttention via the vLLM Python module (Kwon et al., 2023). For sampling answers from the LLM, we use simple autoregressive decoding, only sampling from the tokens with the $50$ highest logits and with a probability of at least $95\%$ with a sampling temperature of $0.1$.

### A.3  LLM OUTPUTS

When first approaching the problem of using an LLM as an evaluator for NetHack, we sought to understand whether the LLama 2 model had observed enough knowledge of NetHack to be useful

---

**What do you know about NetHack and the goal of a player in it? (Llama 2 70b)**

```
NetHack is a popular open-source roguelike video game that was first
released in 1987.  It is a descendant of the 1980 game Rogue, and
it has been widely influential in the development of the roguelike
genre.  The game takes place in a procedurally generated dungeon,
where the player controls a character known as the "hero" who must
navigate through levels filled with monsters, traps, and treasure.

The ultimate goal of the player in NetHack is to find and retrieve
the Amulet of Yendor, which is located on the lowest level of the
dungeon.  However, this task is not straightforward, as the amulet is
guarded by powerful monsters and protected by various obstacles.  To
reach the amulet, the player must explore the dungeon, fight or evade
monsters, collect useful items and equipment, and solve puzzles.
Along the way, the player can also complete side quests and interact
with non-player characters (NPCs) for rewards and assistance.

It's worth mentioning that NetHack is known for its high difficulty
level, complex gameplay mechanics, and steep learning curve.  Players
must carefully manage their resources, such as food, water, and
health, and make strategic decisions about how to proceed through
the dungeon.  Additionally, the game features permadeath, meaning
that if the hero dies, they must start over from the beginning with a
new character.  Despite these challenges, NetHack remains a beloved
game among fans of the roguelike genre due to its depth, complexity,
and replayability.
```

Output 1: Answer of Llama 2 70b (chat) on a question about its knowledge of NetHack. The model demonstrate knowledge of both the history of the game, its goal and peculiarities of its gameplay.

---

**What do you know about NetHack and the goal of a player in it? (FLAN-T5 XXL)**

```
NetHack is a first-person shooter video game
```

Output 2: Answer of FLAN-T5 XXL (Chung et al., 2022) on a question about its knowledge of NetHack. The model only provides an factually incorrect short answer.

at this task. In Output 1, we directly ask the model about its knowledge of NetHack, obtaining a complete answer, showing a remarkable knowledge of NetHack's game dynamics. By contrast, a FLAN-T5 XXL model (Chung et al., 2022) seem not to know anything about NetHack, and hallucinates a brief answer about it, falsely claiming it to be a first-person shooter video game. Future work can explore the possibility to fine-tune an LLM to apply Motif to domains unknown to it.

We additionally provide two examples of responses from Llama 2 70B. In the first example in Output 3 we compare the message "There is a staircase up here. You see here an orcish helm." to "It's solid stone." We notice that the LLM naturally associates the staircase with potential progress in the game, making use of its common sense, knowledge of NetHack and predictive capabilities. In the second example in Output 4 we compare an empty message (which happens often in NetHack) to "It's solid stone." In this case, the information content of both messages is extremely low and the LLM naturally recognizes that neither is better than the other.

---

**"There is a staircase up here. You see here an orcish helm."**
**vs**
**"It's solid stone."**

```
I have extensive knowledge of NetHack and its mechanics.  The goal of
NetHack is to descend through the dungeon levels, collecting treasure
and killing monsters, while avoiding death and reaching the Amulet
of Yendor, which is located on the bottom level.  The game is won by
taking the Amulet to the surface.

Description 1 mentions a staircase up and an orcish helm.  The
presence of a staircase suggests the possibility of descending
deeper into the dungeon, which aligns with the goal of the game.
The orcish helm could potentially provide a bonus to the player's
combat abilities, which would aid in killing monsters and collecting
treasure.

Description 2 simply states that the area is solid stone.  This
does not provide any information about the possibility of descending
deeper into the dungeon or encountering monsters or treasure.

Based on the information provided, Description 1 is more likely
to lead to progress towards the goal of the game.  Therefore,
("best_description":  1).
```

---

Output 3: Answer of Llama 2 70b with default prompt when tasked to evaluate two messages. The model prefers the message indicating progress, containing both reference to a staircase and to one of the objects in the game, to the less informative message "It's solid stone". The anticipatory behavior of the LLM, guided by its knowledge of the task, is clearly visible.

## A.4 ANALYZING MOTIF'S INTRINSIC REWARD

One way to understand which kind of behaviors might be encouraged by Motif is to directly inspect the intrnsic reward function it produces. To do so, we compute the reward for all the messages that can be found in a hold-out set of $\mathcal{D}_{\text{text}}$, and rank the messages by the value of the reward.

In Message List 1, we report a list of the top messages that are preferred by the reward function when trained with the annotations produced by the LLM with the default prompt. As highlighted in Section 4.2, among the most encouraged messages are the ones that are seen by the agent during the execution of exploratory actions, like opening a door, finding a hidden door, or moving a boulder that blocks the way to the rest of the dungeon. We discussed the remarkable effect that this can have on exploration, since the agent does not have to learn that the consequences of this simple and common actions are often positive for getting new information or arriving to a better state of the environment. Other encouraged actions are more directly related to successful accomplishments in the game, like descending the stairs to the next dungeon level, killing or fighting monsters, and seeing potentially useful objects. We also notice some general bias of the LLM to prefer longer messages, which may negatively affect the quality of the resulting reward function.

We additionally report the messages with lowest reward function value in Message List 2. Many of these correspond to undesirable behaviors in the game, such as executing actions in the wrong moment, attacking non-playing characters (like Sneem) or damaging a pet kitten. Overall, our thresholding mechanism will discard the effect of this tail behavior of the reward function, losing semantics in the reward function in order to reduce noise.

## A.5 DIVERSE BEHAVIORS THROUGH DIVERSE PREFERENCES

We now have a closer look at the reward functions originated in the experiment from Table 1.

---

**Empty message
vs
"It's solid stone."**

```
I have extensive knowledge of NetHack and its mechanics.  The goal of
NetHack is to descend through the dungeon levels, collecting treasure
and killing monsters, while avoiding death and reaching the Amulet
of Yendor, which is located on the bottom level.  The game is won by
taking the Amulet to the surface.

Based on the given descriptions, neither of them provides much
information about the game state or the player's progress.

Description 1 is empty, providing no information about the game
state.

Description 2 mentions that the current square is solid stone, which
means that the player is currently standing on a solid stone floor.
This information doesn't reveal much about the player's progress or
the surrounding environment.

Comparing the two descriptions, neither provides any information
about the player's progress towards the goal of the game.

Based on the information provided, it's difficult to determine
which description is more likely to make progress towards the goal.
Therefore, I must declare that ("best_description":  None) is the most
appropriate response.

However, I must note that in general, it's important to keep in
mind that the player's surroundings, inventory, and current floor
level are important factors in determining progress in NetHack.  A
description that provides information about these aspects would be
more helpful in evaluating progress towards the goal.
```

---

Output 4: Answer of Llama 2 70b with default prompt when tasked to evaluate an empty message to the message "It' solid stone", which is encountered by the agent when bumping into a wall. For two equally not helpful messages, the model is able to refuse to give an answer.

For each of the three agents (*The Gold Collector*, *The Descender*, *The Monster Slayer*), we want to measure which messages are getting a relatively higher reward, compared to the baseline reward function created by the most basic version of Prompt 2. To do so without being subject to the confounding factor of potentially different reward scales, we compute, for the same set of messages from the held-out set of $\mathcal{D}_{\text{pref}}$ the ranking of messages according to the different rewards. Then, we report for each one of the steered agents the messages that maximally increased their ranking among the other messages, compared to the rankings observed with the zero-knowledge prompt.

In Message List 3, we see that *The Gold Collector* mainly prefer the collection of gold pieces, or any other actions, even from other characters in the game, related to money. In addition, explorative actions such as interacting with doors (related both to "In what direction" and "The door resists") are encouraged. Again, this is an instance of anticipatory rewards, or the intrinsic reward from the LLM behaving similarly to a value function: the LLM imagines that, after opening a door, there can be some gold awaiting for the agent.

Message List 4 shows that *The Descender*'s reward has an even stronger preference for messages suggesting that exploratory behaviors are being executed, such as interacting with doors, recovering sight, or even swapping places with the pet, that the LLM interprets as a sign of swift movement in the game.

**Highly preferred messages**

```
The door opens.
With great effort you move the boulder.
You descend the stairs.
You find a hidden door.
You kill the cave spider!
As you kick the door, it crashes open!
You kill the newt!
You kill the grid bug!
You find a hidden passage.
You see here a runed dagger.
You hear the footsteps of a guard on patrol.  It's a wall.
There is a partly eaten rothe corpse here; eat it?  [ynq] (n)
There is a sewer rat corpse here; eat it?  [ynq] (n)
You hit the gnome lord!
A kobold lord blocks your path.
You kill the gecko!  Welcome to experience level 3.  You feel
healthy!
You hit the black naga hatchling!  The black naga hatchling bites!
$ - 3 gold pieces.
```

Message List 1: Most preferred messages according to the reward function induced by feeding the default prompt to the LLM for annotating the dataset.

**Least preferred messages**

```
Unknown command 'ⁿM'.
Thump!
This time I shall let thee off with a spanking, but let it not happen
again.
The little dog jumps, nimbly evading your kick.
You swap places with your kitten.
It yowls.
You miss it.
You kick the kitten.  The kitten jumps, nimbly evading your kick.
This orange is delicious!  You feel weak now.
It hits!
Really attack Sneem?  [yn] (n)
You collapse under your load.  It is hit.
Core dumped.
You hit it.  You are beginning to feel hungry.
Everything looks SO boring now.
A glowing potion is too hot to drink.
```

Message List 2: Least preferred messages according to the reward function induced by feeding the default prompt to the LLM for annotating the dataset.

Lastly, in Message List 5 we report the relatively most preferred messages for *The Monster Slayer* prompt. As expected, all the messages that are more preferred compared to the baseline are related to combat. Interestingly, when pushed to do so, the reward function starts to encourage attacking the pet, which was among the least encouraged messages for the default prompt. This comes from the prompt, the generally prescribes combat as the goal for the agent to pursue, regardless of the target.

---

**Relatively most preferred messages for *The Gold Collector***

```
In what direction?
$ - 2 gold pieces.
$ - 4 gold pieces.
$ - 5 gold pieces.
You hear someone counting money.
$ - 7 gold pieces.
$ - 3 gold pieces.
You hear someone counting money.  It's solid stone.
You see here a tin.
The door resists!
```

---

Message List 3: Messages more emphasized by the prompt used by *The Gold Collector* compared to the baseline prompt.

---

**Relatively most preferred messages for *The Descender***

```
In what direction?
The door resists!
You can see again.
You see here a whistle.
You are lucky!  Full moon tonight.
You swap places with your little dog.
You see here a tin.
You see here a gnome corpse.
You swap places with your kitten.
You find a hidden door.
```

---

Message List 4: Messages more emphasized by the prompt used by *The Descender* compared to the baseline prompt.

## A.6 ADDITIONAL BASELINES

For the sake of clarity, in the main paper we have only compared to the RND baseline (Burda et al., 2019b). We now provide a comprehensive comparison to more baselines, including additional intrinsic motivation baselines and ablations on our method.

In our first set of baselines we compare to the Exploration via Elliptical Episodic Bonuses (**E3B**) baseline (Henaff et al., 2022) which has shown state-of-the-art performance on the MiniHack game (Samvelyan et al., 2021). E3B leverages an inverse dynamics model to learn the important features on which they define episodic elliptical bonuses. We also compare to **NovelD** (Zhang et al., 2021) which proposes a measure of novelty inspired by RND and makes use of episodic counts. NovelD previously claimed state-of-the-art performance on some of the NetHack environments. We additionally perform an ablation our method where we do not leverage LLM feedback to define a reward function and instead assign a value of 1 to all messages. We call this baseline **Motif w/o LLM**. Finally, we considered implementing Behaviour Cloning from Observation (**BCO**) (Torabi et al., 2018), however this baseline typically never outperforms the agent from which it learns. As we notice in our experiments, Motif signficantly improves upon the agent that generated the initial dataset, sometimes doubling the base performance (see Appendix A.8.3)

We additionally compare to two recent LLM-based baselines. The first baseline we investigate is the **LLM-as-a-policy** approach of recent work Wang et al. (2023), however the agent would not make any progress. It is possible that with much more prompt engineering this result could improve, however we believe the appeal of Motif is precisely in avoiding such involved optimization.

**Relatively most preferred messages for *The Monster Slayer***

```
You hit the newt.
You miss the newt.
You see here a jackal corpse.
Really attack the little dog?  [yn] (n)
You hit the lichen.
The goblin hits!
The giant rat bites!
The grid bug bites!
You see here a newt corpse.
The jackal bites!
```

Message List 5: Messages more emphasized by the prompt used by *The Monster Slayer* compared to the baseline prompt.

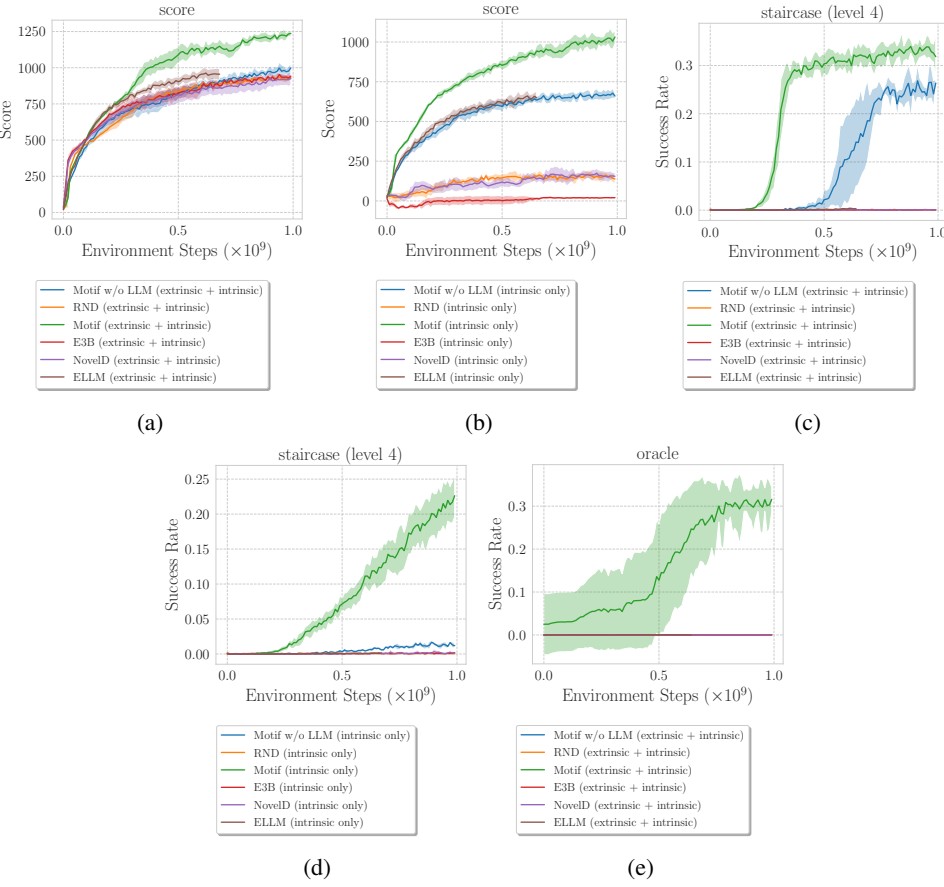

Figure 7: Comparison to additional baselines on the `score` task, the `staircase (level 4)` task and on the `oracle` task, both when learning only from intrinsic reward and when it is combined with extrinsic.

Indeed, Motif's prompts are simple and could be composed by non-experts. The second LLM-based baseline we implemented is a version of the recently-proposed ELLM algorithm (Du et al., 2023). Our implementation of ELLM closely follows the details from the paper. We use the cosine similarity between the representation, provided by a BERT sentence encoder (specifically the

same `paraphrase-MiniLM-L3-v2` model), of the game messages and the "early game goal" extracted from the NetHack Wiki as intrinsic reward. These first two lines are taken from the early stategy section and go as follows "The goals of the early game are completing the first two branches off the main dungeon — Sokoban and the Gnomish Mines — and gathering basic supplies and re-sistances. You may find some nice armor or sacrifice for a shiny artifact weapon, but such luck does not always happen.". Notice that Motif does not use this kind of privileged information.

In Figure 7 we compare Motif to the aforementioned baselines in both the intrinsic only setting and when learning from the intrinsic and extrinsic rewards. We make this comparison in the standard `score` task, the `staircase (level 4)` task and the `oracle` task.

We see that in the `score` task, Motif easily outperforms existing intrinsic motivation approaches and LLM-based methods. In fact, neither the intrinsic motivation baselines or the LLM-based methods are significantly improving upon the extrinsic-only agent. On the more straightforward `staircase (level 4)` task, some of the baselines provide gains, although they require access to the extrinsic reward to do so. None of the baseline approaches showed any meaningful progress in the environment `oracle`, characterized by extremely sparse rewards.

For each baseline we have performed a sweep over their hyperparameters, in particular the intrinsic rewards coefficients. We kept the extrinsic rewards coefficients fixed (1.0 for `score` and 10.0 for the rest). For E3B, we swept the ridge regularizer in the values $\{0.1, 1.0\}$ and the intrinsic reward coefficient in the values $\{0.0001, 0.001, 0.01, 0.1, 1.0, 10.0\}$. The final values were 0.1 and 0.1, re-spectively. For NovelD, we swept the scaling factor in $\{0.1, 0.5\}$ and the intrinsic reward coefficient in the values $\{0.0001, 0.001, 0.01, 0.1, 1.0, 10.0\}$. The final values were 0.5 and 0.1, respectively. For RND, we swept the intrinsic reward coefficient in the values $\{0.001, 0.01, 0.1, 0.5, 1.0, 10.0\}$. The final value was 0.5. For the ELLM baseline, we swept the value of the intrinsic coefficient parameter from the values $\{0.01, 0.05, 0.1, 1.0\}$. The final value was 0.1.

Finally, to further assess the importance of using a strong LLM model for good performance, we additionally compare to a base-line that builds on a sentiment analysis model, in particular the `t5-base-finetuned-imdb-sentiment` model. This base-line closely follows the Motif algorithm but replaces the Llama 2 model with this sentiment analysis model. More specifically, we extract, for each message, a score computed as the sigmoid of the confidence of the model in its positive or negative sentiment predic-tion. Then, for each pair of messages, we assign a preference based on the message with higher sentiment score. Finally, we execute reward training and RL training as with Motif. We present results in Figure 8 for the `score` task where we see that its performance is near zero, both with and without extrinsic reward. This poor perfor-mance can be easily explained: a generic sentiment analysis model cannot capture the positivity or negativity of NetHack-specific cap-tions. For instance, killing or hitting are generally regarded as neg-

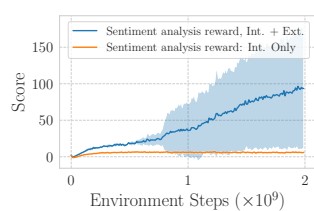

Figure 8: Performance of a baseline using a sentiment analysis model instead of the Llama 2 LLM in order to ex-tract preferences over pairs of messages.

ative statements, but they become positive in the context of killing or hitting monsters in NetHack. Llama 2 can understand this out-of-the-box without any fine-tuning, as demonstrated by our experi-ments.

### A.7 EXPERIMENTAL DETAILS

We base our implementation of the environments on the code released in the NetHack Learning Environment (Küttler et al., 2020). We use default parameters for each environment. However, as discussed in one the issues currently available on the public repository, even tough the 'eat' action is available to the agent, it is not possible to actually eat most of the items in the agent's inventory. To overcome this limitation, we make a simple modification to the environment by letting the agent eat any of its items, at random, by performing a particular action (the action associated with the key `y`). This effectively addresses this mismatch. Additionally, for our experiments on learning from intrinsic rewards only we let the agent interact with the environment even if it has reached the goal (although we only reward it once). This was also noted in (Burda et al., 2019b) in their

intrinsic-only experiments, although they make this adjustment on the level of the agent itself by never experiencing termination (e.g. $\gamma = 0.99$ for all states).

During the reward training phase of Motif, we use the message encoder from the Elliptical Bonus baseline (Henaff et al., 2022). This baseline has shown state of the art performance on MiniHack (Samvelyan et al., 2021). We split the dataset of annotation into a training set containing $80\%$ of the datapoints and a validation set containing $20\%$. We train for 10 epochs, around which time the validation loss stabilizes. We use a learning rate of $1 \times 10^{-5}$.

Before providing the reward function to the RL agent, we normalize it by subtracting the mean and dividing by the standard deviation. As Equation 2 shows, we further divide the reward by an episodic count and we only keep values above a certain threshold. The value of the count exponent was 3 whereas for the threshold we used the $50th$ quantile of the empirical reward distribution.

Table 2: PPO hyperparameters

| Hyperparameter | Value |
|---|---|
| Reward Scale | 0.1 |
| Observation Scale | 255 |
| Num. of Workers | 24 |
| Batch Size | 4096 |
| Num. of Environments per Worker | 20 |
| PPO Clip Ratio | 0.1 |
| PPO Clip Value | 1.0 |
| PPO Epochs | 1 |
| Max Grad Norm | 4.0 |
| Value Loss Coeff | 0.5 |
| Exploration Loss | entropy |
| Extrinsic Reward Coeff (`score`) | 0.1 |
| Extrinsic Reward Coeff (others) | 10.0 |
| Intrinsic Reward Coeff | 0.1 |
| $\epsilon$ threshold | 0.5 quantile |
| $\beta$ exponent | 3 |

For the RL agent baseline we build on the Chaotic Dwarven GPT-5 baseline (Miffyli, 2022), which itself was defined on Sample Factory (Petrenko et al., 2020). Sample Factory includes a an extremely fast implementation of PPO (Schulman et al., 2017) which runs at about $20K$ frames-per-second using 20 computer cores and one $V100$ GPU. We provide all hyperparemeters in Table 2.

For the experiments in Section 4.1, all results are reported by averaging 10 random seeds together with the standard deviation. For the experiments in Section 4.4, report results by averaging 5 random seeds together with the 95th confidence interval.

## A.8 ADDITIONAL EXPERIMENTS AND ABLATIONS

### A.8.1 ADDITIONAL EXPERIMENTS

**Un-hackable oracle task** In Section 4.2, we have investigated the behavior of the agent on a range of task and our analysis has revealed an unexpected way in which the agent solves the `oracle` task.

This begs the question whether Motif could solve the intended task, that is, to go down multiple dungeons and find the oracle in the right branch of the maze. To do so, we modify the `oracle` to include the following success condition: the task is done when the agent stands by the oracle and is not under a state of hallucination. We name this task `oracle-sober`.

In Figure 9 we show that Motif, using the intrinsic and extrinsic rewards, is still able to solve this extremely sparse reward task, although with a lower success rate than before. When using only the intrinsic reward Motif performs slightly better than the baselines. It would be possible to improve this performance by explicitly mentioning the task of interest in the prompt, similarly to results presented in Section 4.4. In particular, the oracle only appears in one of two

branches going down the NetHack dungeons. By modifying the prompt we could encourage the agent to visit the right branch and possibly significantly increase its chance of finding the oracle.

**Alignment in avoiding pet killing** The agent is accompanied by a pet from the beginning of the game, that can help it to kill monsters or pick objects. Following the environment reward, there is a strong incentive to kill the pet (the agent trained with extrinsic reward kills it $99.4\% \pm 0.63\%$ of the time), since the agent gets score points by doing it and additionally avoids to lose points due to future actions of the pet (e.g., when the pet kills monsters in place of the agent). This behavior is, however, not intuitive for most humans, that would not kill the pet unless constrained by the game to do so. Indeed, Motif's intrinsic reward captures this intuition, and, despite achieving a better score, an agent trained with that reward kills the pet significantly less, $33.4\% \pm 25.14\%$ of the time.

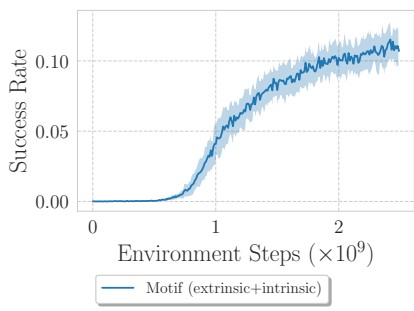

Figure 9: Performance of Motif on the unhackable version of the `oracle` task. Motif reaches satisfying performance even in this case.

### A.8.2 ABLATIONS

In the design of Motif we we introduce two hyperparameters shown explicitly in the main paper in Section 3. These hyperparameters consist in the exponent $\beta$ that affects the counts coefficient and the threshold value $\epsilon$ under which rewards are zeroed out. We verify the effect of these hyperparameters on the `score` task and present results in Figure 10. We notice that Motif's performance is generally robust across a wide range of values. An interesting failure case is when there are no counts (i.e. when $\beta = 0$). This is due to the fact that the LLM encourages the agent to try to interact with different objects in the game, such as armor and weapons, in order to increase its abilities. This is a standard strategy, however, in the current version of `score` the action set does not include the possibility to interact with such objects, which brings the agent to never-ending loops where it seeks things it simply cannot achieve.

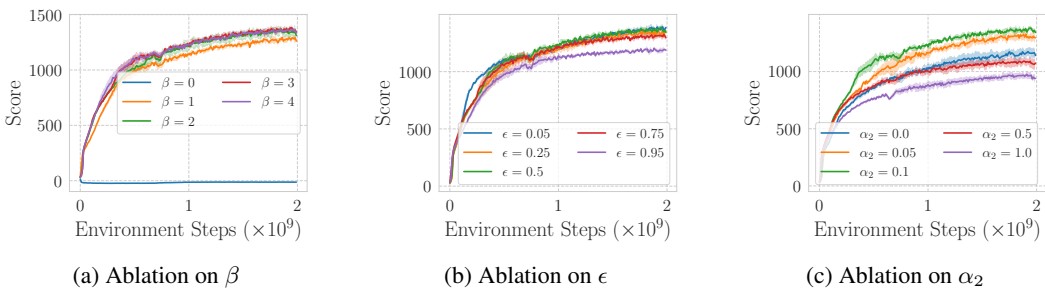

(a) Ablation on $\beta$  (b) Ablation on $\epsilon$  (c) Ablation on $\alpha_2$

Figure 10: Ablations on the hyperparameters used by Motif. Using a form of count-based normalization is critical for the method to work. Otherwise, Motif is very robust to hyperparameter choices and the change in performance follows intuitive patterns.

Finally, another important hyperparameter is the set of coefficients that balance the intrinsic and extrinsic rewards. As we have seen previously, learning only with the intrinsic reward leads improved performance when compared to an agent learning with the extrinsic reward. As such, we vary the value of the extrinsic reward coefficient and present results in Figure 10c. We notice that when the extrinsic reward coefficient is in the same range as the intrinsic reward coefficient (that is a value of around $0.1$), we achieve the best performance. As we increase the value of the extrinsic reward coefficient, the performance tends to decrease, eventually reaching the same score as the extrinsic only agent.

In Figure 6a we have reported the scaling profile when modifying the LLM annotator size on the `staircase (level 3)`. We now additionally investigate the effect of the model size on the

`oracle` task and present results in Figure 11a. We notice that the 70B model finds much more consistently the oracle than both the 13B and the 7B ones.

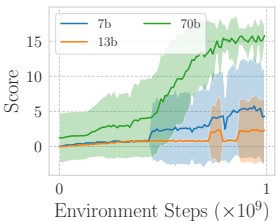
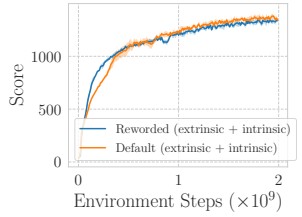
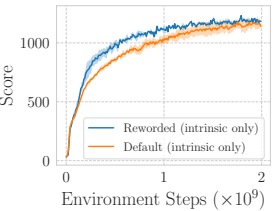

(a) Performance with different model sizes on `oracle`.

(b) Performance on the `score` task (extrinsic + intrinsic).

(c) Performance on the `score` task (intrinsic only).

Figure 11: Additional experiments on sensitivity of the performance to changes in the LLM and prompt. Good performance on the `oracle` task emerges from the 70b Llama 2 model, and performance on the `score` task is reasonably robust to a rewording of the prompt, both with and without extrinsic reward.

We also previously investigated in Figure 6c the importance of the precise prompt used to obtain preferences from the LLM and noticed significant differences when the task has a considerably narrow definition of success as is the case in `oracle`. We now investigate whether this also appears to be case in the more open-ended `score` domain. In Figure 11b we plot the learning curves when the agent learns through both the extrinsic and the intrinsic reward functions, whereas in Figure 11c we report results when only learning from the intrinsic reward function. We notice no significant difference in performance in this case between the default prompt and the reworded one. This indicates that when the task can be achieve by a larger span of behavior, the performance obtained by the RL agent is robust to variance the prompt .

### A.8.3 IMPACT OF DATASET DIVERSITY AND PERFORMANCE LEVEL

In all of our experiments, we use a dataset $\mathcal{D}$ collected by policies trained with RL. From the base dataset, we extracted a 500000-pairs dataset and annotated it using an LLM's preferences. How does the performance of Motif changes based on the return level and diversity of the policies that collected the dataset? This question is related to recent studies conducted in the context of offline RL (Lambert et al., 2022; Schweighofer et al., 2022), that similarly identified the performance level and diversity of a dataset as the important features to measure how good will a resulting policy be after running offline RL training on the dataset.

To control the diversity and performance level in the dataset, we characterize polices that collected the dataset using the distributions of their returns (Rahn et al., 2023). In particular, for each one of the pairs of observations in the dataset, we use the game score of the episode from which an observation came from. Figure 12 shows the distribution of such game scores. The histogram shows

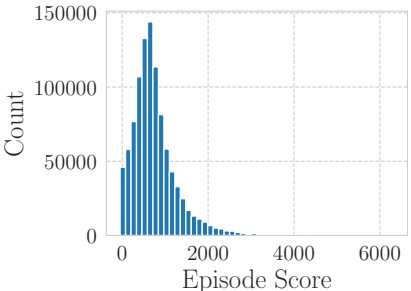

Figure 12: Distribution of scores of the episodes from which the observations in the dataset of pairs $\mathcal{D}_{\mathbf{pref}}$ come from. The distribution exhibits a long right tail, with agents that, due to lucky configurations of the procedurally generated dungeon, can sometimes get particularly high scores.

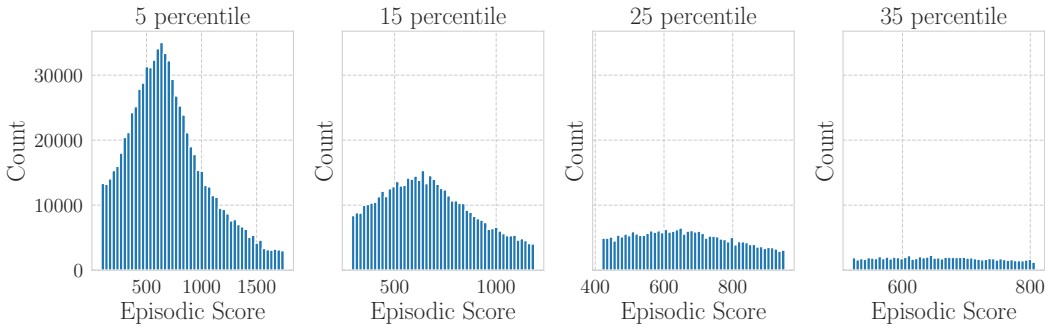

Figure 13: Distribution of scores of the episodes from which the observations in the dataset of pairs $\mathcal{D}_{\mathbf{pref}}$ come from, cut (from above and below) at different percentile levels $\eta$. This progressively reduces the overall diversity in the dataset.

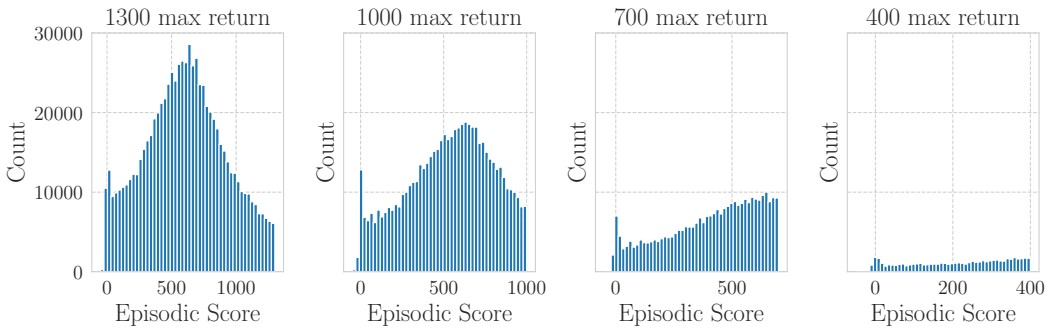

Figure 14: Distribution of scores of the episodes from which the observations in the dataset of pairs $\mathcal{D}_{\mathbf{pref}}$ come from, discarding pairs that contain at least one observation coming from an episode with a level of score greater than the maximum return threshold. This progressively reduces the performance of the policies that collected the dataset.

a reasonably diverse score distribution, highlighting a certain degree of diversity that comes both from the variability across different seeds of the RL algorithm and the variability across episodes due to the procedurally-generated nature of NetHack.

To see how changes to the distributions of scores, and thus to the diversity and the performance level of the dataset, have an impact on the reward function and the downstream performance of Motif, we alter the dataset in two controlled ways. First, to reduce the diversity of the dataset, we remove the left and right tails of the distribution, by discarding pairs that contain at least an observation coming from episodes with score residing in either one of those tails. We identify those tails by measuring what is lower or higher than a given percentile, namely 5%, 15%, 25%, 35% for the left tail, and 95%, 85%, 75%, 65% for the right tail respectively. We denote by $\eta$ this "symmetric" percentile level. In Figure 13, we show the resulting distribution of scores for different percentile levels. Second, to reduce the maximum performance implied by the dataset, we remove the pairs containing observations coming from episodes that achieved a at least a given level of return. In Figure 14, we show how the distribution changes for different levels of score-based filtering, with maximum returns of 1300, 1000, 700, 400.

We report in Figure 15 the performance of Motif when using reward functions derived from the dataset restricted in the two different ways. The results show that Motif is remarkably robust to both the performance level and the diversity of the dataset. Up to a maximum score of 700 and a percentile of $\eta = 25\%$, corresponding to dataset sizes of around 100000 samples, Motif's experiences only minimal drops to its performance. The performance completely degrades only for extreme removals from the dataset, shrinking it down to a few tens of thousands of pairs (i.e., for $\eta = 35\%$ and a maximum score of 400).

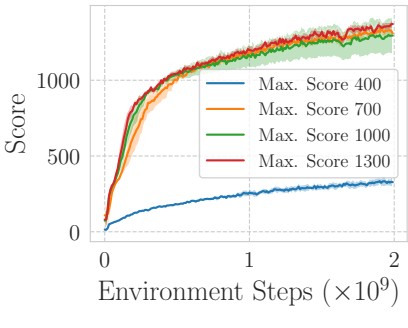

(a) Performance-based restriction

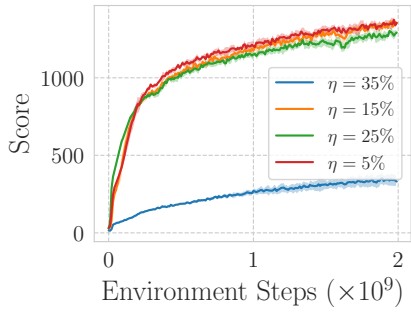

(b) Diversity-based restriction

Figure 15: Performance of Motif trained with a combination of intrinsic and extrinsic reward, when trained the reward on a restricted dataset of preferences, either capping the score of the best policy in the dataset (a) or reducing its diversity by discarding observations coming from episodes with score higher or lower than the one at a given reference percentile $\eta$ (b).

### A.8.4 COMPUTATIONAL CONSIDERATIONS

We now provide information regarding the computational costs for reproducing Motif's results. Our algorithm was purposefully constructed in a modular fashion where each of the phases (dataset annotation, reward training and RL training) can be ran independently of the others.

Annotating a dataset of 500k pairs of captions on eight V100s GPU takes at most 6 GPU days when using the Llama 2 7B model, 8 GPU days when using the 13B model and 15 GPU days when using the 70B model. If annotation is done on A100s GPUs the compute costs can be cut approximately in half. Additionally, our code allows running this experiment asynchronously over a greater number of GPUs, which can significantly reduce the annotation time. In our experience, this can take as little as 6 hours on 10 nodes with eight V100s each.

To further encourage reproducibility and scientific discoveries, we also release our complete Llama 2 annotations for all experiments. This means that researchers can directly build on them and focus on either the reward training and the RL training phases.

Training the reward model requires only about 1 hour on one V100 GPU, while the RL training loop takes about 24 hours on one V100 GPU.

