# OpenReview forum: "Motif: Intrinsic Motivation from Artificial Intelligence Feedback"
_ICLR.cc/2024/Conference — ICLR 2024 poster_

### Official Review · Reviewer_j8Na · 2023-10-29

**Soundness:** 2 fair
**Presentation:** 3 good
**Contribution:** 2 fair
**Rating:** 5
**Confidence:** 4

**Summary:**

This paper proposes Motif: 1) replacing human labeling in preference-based RL with LLM labeling, and 2) joint optimization of preference-based and extrinsic rewards to solve the NetHack environment. After collecting sufficiently covered offline data from the existing RL methods, preference labels are annotated using LLaMA 2. Motif trains the preference reward model from those data and leverages it for online training from scratch, jointly maximizing preference-based and extrinsic rewards. Motif exhibits strong performance in staircase tasks from NetHack.

**Strengths:**

### quality and clarity
- This paper is well-written and easy to follow.

### significance
- The empirical results are strong. It would be notable to solve the difficult, sparse-reward NetHack environments that previous intrinsic-motivation methods cannot solve by leveraging preference-based reward.

**Weaknesses:**

- I think the point of this paper is that "joint optimization of preference-based and extrinsic reward helps resolve the sparse reward problems". As the source of feedback, either humans or LLMs are OK. I think describing this as LLM's contribution might be an overstatement.
- As a preference-based RL method, I guess there are no differences from the original paper [1]. In the LLM literature, [2] leverages GPT-4 to solve game environments, and [3] incorporates LLM-based rewards for RL pretraining.
- Terminology: I'm not sure if a preference-based reward should be treated as an "intrinsic" reward. I think it is extrinsic knowledge (from humans or LLM).

[1] https://arxiv.org/abs/1706.03741

[2] https://arxiv.org/abs/2305.16291

[3] https://arxiv.org/abs/2302.06692

**Questions:**

- Which RL algorithm is used for Motif? I may miss the description in the main text.
- Are there any reason why employ LLaMA 2 rather than GPT-3.5 / 4?

(Minor Issue)
- In Section 2, `... O the observation space ...` might be `... O is the observation space ...`.

---

> ### Author Response · Authors · 2023-11-15
>
> Thank you for your feedback!
>
> > I think the point of this paper is that "joint optimization of preference-based and extrinsic reward helps resolve the sparse reward problems". As the source of feedback, either humans or LLMs are OK. I think describing this as LLM's contribution might be an overstatement. [...] As a preference-based RL method, I guess there are no differences from the original paper [1].
>
> We never claim that an LLM's feedback is inherently better than the feedback coming from humans, even though we believe assessing whether that could be the case is an interesting avenue for future work. Instead, we simply leverage an LLM's feedback because of its scalability: in just a few hours on a small-sized cluster, one can annotate thousands of pairs of games events, which would require significant amounts of human NetHack experts's labour otherwise. This scalability, leveraged also in recent work on chat agents (e.g., Constitutional AI from Bai et al, 2022), allows for a method like Motif to fully leverage human common sense to bootstrap an agent's knowledge.
>
> > In the LLM literature, [2] leverages GPT-4 to solve game environments, and [3] incorporates LLM-based rewards for RL pretraining.
>
> In [2] the Voyager algorithm uses complex prompt engineering involving significant amounts of human knowledge and engineering (such as deliberately prompting the LLM to use the “explore” command regularly). Additionally, Voyager relies on a Javascript API to bridge the gap between the LLM's language outputs and the high dimensional observations and actions of Minecraft. Finally, Voyager relies on perfect state information about certain features of the game (e.g. agent position and neighbouring objects). Voyager also builds on GPT-4's strong coding abilities, which would likely not be the case of current open models. Altogether, these factors strongly limit Voyager's general applicability and reproducibility. On the other hand, Motif relies on very limited human knowledge, being able to get significant performance even without any information about NetHack. Moreover Motif is way simpler to implement, with very few, clearly separated, moving pieces, providing a robust solution for leveraging prior knowledge from large models. This makes our approach a significantly more general method that has the potential to be applied to multiple domains or be possibly combined with powerful Large Vision Language Models.
>
> We have additionally compared Motif to the ELLM approach of [3], adapted as described in the general response, showing that Motif significantly outperforms it in all NLE tasks. Please see the common response for a detailed description of the experiment. The paper also highlights in Section 5 (Related Work) the important differences between Motif and the ELLM algorithm. In particular, ELLM's reward function cannot, by design, exhibit the exploration and credit assignment properties of the one produced by Motif (see Section 4.2 of the paper, "Alignment with human intuition"). We believe those differences are key to the strong performance of Motif.
>
> > Terminology: I'm not sure if a preference-based reward should be treated as an "intrinsic" reward. I think it is extrinsic knowledge (from humans or LLM).
>
> As standard, we refer to extrinsic reward as the reward that comes from the task to be performed in the environment, whereas intrinsic rewards are provided by the algorithm. From that point of view, the reward provided by the LLM is intrinsic (as opposed to extrinsic) to the agent. Please notice that this terminology has previously been used in the literature (see, for instance, the ELLM [3] paper).
>
>
> > Which RL algorithm is used for Motif? I may miss the description in the main text.
>
>
> We use the asynchronous PPO implementation of Sample Factory (Petrenko et al., 2020). This information is available in the paper on the bottom of page 4. We chose this implementation as it extremely fast: we can train an agent on 2B steps of interactions in only 24h using only one V100 GPU.
>
>
> > Are there any reason why employ LLaMA 2 rather than GPT-3.5 / 4?
>
> Yes, we believe there are important and significant reasons to prefer using Llama 2 rather than GPT3.5 or GPT4. GPT3.5 and GPT4 are subject to changes over time, require significant financial efforts to be used at scale, and rely on unknown methodologies and practices. Despite the fact that they might provide better performance, they are problematic for rigorous scientific reproducibility, and thus they are significantly less preferrable than Llama 2 for conducting scientific research. We explicitly made this decision for the benefit of the scientific community, and we will also release our code and dataset to ease experimenting with a method like Motif for other members of the community.
>
> > (Minor Issue)
>
> We thank the reviewer for spotting the typo. We corrected it in the updated version of the paper.

---

> ### Author Response · Authors · 2023-11-17
>
> Dear reviewer,
>
> We believe our response addresses your concerns. In particular, we have now a total of 7 different baselines, including two additional baselines using language model-based rewards (sentiment analysis and ELLM), which are all significantly outperformed by Motif.
>
> If there are any remaining concerns which prevent you from increasing your score, please let us know in the discussion as soon as possible and we will do our best to address them.
>
> Thank you,
>
> The Authors

---

> ### Comment · Reviewer_j8Na · 2023-11-18
>
> Thank you for the detailed response.
>
> My questions on (1) the difference between yours and voyager/ELLM, (2) the RL algorithm, and (3) the choice of LLMs become clear now. Here are the remaining comments:
>
> **Contribution:**
>
> The response to my concern seems to be reasonable, but not explicitly stated in the main text. So, the reader should infer the hidden intention why LLM is chosen rather than humans as the preference labeler (though alignment with *human* intention is emphasized many times). It is necessary to include your explanation (the scalability of LLM allows Motif to fully leverage human common sense to bootstrap an agent's knowledge, etc.) into the paper in this revision phase.
>
> **Terminology:**
>
> I cannot agree with your explanation because, in deep RL literature, the preference-based reward should be distinguished from "intrinsic motivation".  For instance, The strong performance of Motif with "intrinsic reward only" in Figure 1 causes confusion as to why the agent can learn decent behavior only with "intrinsic motivation",  since "intrinsic reward" often reminds the community of exploration. However, if the reader understands that Motif learns preference-based reward as a proxy for extrinsic reward, the results in Figure 1 will make sense. I express strong concerns about the description that equates "intrinsic reward" with a preference-based reward that should be a proxy for extrinsic reward and be learned from external knowledge.

---

> > ### Author Response · Authors · 2023-11-18
> >
> > Thank you for your timely answer! We are glad to know that we have effectively addressed the majority of the reviewer's concerns. We now provide answers to the remaining two concerns on Contribution and Terminology.
> >
> > __Contribution__
> > We appreciate the suggestion from the reviewer. We highlight that we already discuss this early in the paper, in our introduction, stating that "since the idea of manually providing this knowledge on a per-task basis does not scale, we ask: what if we could harness the collective high-level knowledge humanity has recorded on the Internet to endow agents with similar common sense?". To make the advantage of AI feedback even more explicit to a reader, we added a brief but precise sentence to the conclusion, saying that "[Motif] bootstraps an agent's knowledge using AI feedback as a scalable proxy for human common sense.".
> >
> > __Terminology__
> > In our paper, we demonstrate that the reward from Motif is not _only_ "a proxy for extrinsic reward", but instead captures rich information about the future that helps the agent (1) *explore* unknown parts of the environment, (2) *discover* inherently interesting patterns in the environment and (3) achieve *creative* solutions (please see Section 4.2 "Misalignment by composition in the oracle task"). Notice that these three characteristics are part of the formal definition of intrinsic motivation of Schmidhuber, 1990. In particular, Motif's intrinsic reward "is something that is independent of external reward, although it may sometimes help to accelerate the latter" (Section V.B from Schmidhuber, 1990). Even though we not follow the specific way in which intrinsic motivation is defined in that seminal work (i.e. through learning progress), we believe Motif adhere's to the underlying principles.
> >
> > To better explain what we mean, we report here an excerpt of our response to VWkY, which provides a more detailed discussion on why Motif's reward is much more than simply a replication of the extrinsic reward, and instead incorporates strong elements of intrinsic motivation:
> >
> > > Our paper already provides some insights in the "Alignment with human intuition" paragraph of Section 4.2, but we will now provide an additional perspective that can be beneficial to understand this result. By inspecting the messages preferred by the intrinsic reward function, one can quickly realize that the agent will receive from the LLM's feedback three kinds of rewards: direct rewards, anticipatory rewards and exploration-directed rewards. Direct rewards (e.g., for "You kill the cave spider!") leverage the LLM's knowledge of NetHack, implying a reward very similar to the score (i.e., the extrinsic reward). Motif's reward, however, goes beyond this. Anticipatory rewards (e.g., for "You hit the the cave spider!") implicitly transport credit from the future to the past, encouraging events not rewarded by the extrinsic signal and easing credit assignment. Finally, exploration-directed rewards (e.g., for "You find a hidden door.") directly encourage explorative actions that will lead the agent to discover information in the environment.
> >
> > In this passage we explicitly distinguish the three ways in which Motif helps: (1) through rewards directly related to the score, (2) through anticipatory rewards and (3) through exploration-directed rewards. Notice that if Motif only provided rewards of the type (1), we could see Motif as a proxy to the score. However, (2) and (3) make it abundantly clear that Motif goes much further than that and provides intrinsic motivation to the agent to discover the environment. It is also through (2) and (3) that Motif's performance significantly outperforms the baselines.
> >
> > Finally, we would like to highlight that we are as explicit as we can be as to the nature of the intrinsic reward obtained by Motif, i.e. that it is preference based. This is present in the title, the introduction and throughout the paper at numerous occasions. Notice that it is also the basis for the name of our algorithm (**Motif** -> **Moti**vation from AI **f**eedback).
> >
> > We are hopeful that these answers should address your concerns, but let us know if any further clarification is required.
> >
> > Jurgen Schmidhuber, Formal Theory of Creativity, Fun, and Intrinsic Motivation (1990-2010)

---

> ### Author Response · Authors · 2023-11-22
> **Request for re-evaluation**
>
> Dear reviewer,
>
> We hope our last answer addressed your remaining comments. Let us know if you have any remaining concerns, or if you could now recommend our paper for acceptance.
>
> Best,
>
> The Authors

---

### Official Review · Reviewer_2GW2 · 2023-10-30

**Soundness:** 3 good
**Presentation:** 3 good
**Contribution:** 4 excellent
**Rating:** 8
**Confidence:** 3

**Summary:**

This paper provides Motif, a method for training a reinforcement learning (RL) agent with AI preferences. The main idea of Motif is to use AI preferences (or Large Language Model (LLM) preferences), instead of human preferences, for the preference-based RL. More specifically, Motif trains an intrinsic reward model on a LLM preference dataset, and then trains a RL agent by harnessing the intrinsic reward model. In summary, Motif consists of three phases: (1) dataset annotation by a LLM, (2) reward training on a LLM preference dataset, and (3) RL training with the reward model. This paper applies Motif to the NetHack Learning Environment (NLE). The paper uses Llama-2-70B as a preference annotator, and CDGPT5 as a baseline NetHack agent. This paper shows that RL agents trained with Motif's intrinsic reward surprisingly outperform agents trained using the score itself.

**Strengths:**

- S1. First of all, this paper is well-written and well-organized.
- S2. The idea of using a LLM as a preference annotator for preference-based RL is interesting and promising.
- S3. This paper provides a loss function (equation 1) to train an intrinsic reward model.
- S4. This paper shows that training agents with intrinsic rewards is very effective.

**Weaknesses:**

- W1. One of my main questions is whether Motif can be generally applied to other environments. Even though the NetHack Learning Environment (NLE) is a very challenging environment, it seems that the NLE may be one of environments that a LLM can easily annotate preferences.

**Questions:**

- Q1. Can Motif be applied to other environments beyond the NetHack Learning Environment (NLE)?
- Q2. What a RL algorithm is used for RL fine-tuning?

---

> ### Author Response · Authors · 2023-11-15
>
> Thank you for your feedback!
>
> >  Can Motif be applied to other environments beyond the NetHack Learning Environment (NLE)?
>
> We could not investigate this question in our current paper, as it is already compact with detailed analysis on the behavior, the risks of using LLMs and the possibilities for defining diverse rewards. By adding other environments, we could not provide such in-depth analysis.
>
> We strongly believe that Motif is a general method, and it can applied to other environments after reasonably-sized efforts, when its main assumptions are satisfied. In particular, Motif's LLM needs to have enough knowledge about the environment, which is related to the presence on the Internet of text related to it, and the availability of an event captioning system. These assumptions are realistic in many environments, both when dealing with a physical system (e.g., a robot accompanied by a vision captioner) and a simulated/virtual world (e.g., a commonly-played videogames or Web browsing). Additionally, one could apply the general architecture of Motif to any environments based on visual observations, by just substituting a VLM in place of the LLM. We believe this is an exciting direction for future work.
>
> To give some context on our choice of environment, NetHack is a challenging and illustrative domain to deploy an algorithm like Motif. Captions in NetHack are non-descriptive: they do not provide a complete picture on the underlying state of the game. Moreover, these captions are sparse, appearing in only 20% of states. This means that overall there is a high degree of partial observability. Despite this challenge Motif is able to thrive and show results that we have not witnessed in the literature previously.
>
> We believe that if we were to apply Motif in other environments with more complete descriptions we could see even stronger performance. This would bring important questions to be studied: what exactly is the impact of partial observability on preferences obtained from an LLM? Do more detailed captions unlock increasingly more refined behaviors from the RL agent? Such important questions could be investigated by future work.
>
> > What a RL algorithm is used for RL fine-tuning?
>
> We use the asynchronous PPO implementation of Sample Factory (Petrenko et al., 2020). This information is available in the paper on the bottom of page 4. We chose this implementation as it extremely fast: we can train an agent on 2B steps of interactions in only 24h using only one V100 GPU.

---

> > ### Comment · Reviewer_2GW2 · 2023-11-23
> >
> > Thank you for providing thoughtful responses to my questions. It helped me to understand this work more concretely.

---

### Official Review · Reviewer_VWkY · 2023-10-31

**Soundness:** 3 good
**Presentation:** 4 excellent
**Contribution:** 2 fair
**Rating:** 8
**Confidence:** 3

**Summary:**

The paper introduces Motif, a method for integrating the common sense and high-level knowledge of LLMs into reinforcement learning agents. Motif works by eliciting preferences from the LLM based on pairs of event captions. These preferences are then translated into intrinsic rewards for training agents. The authors test Motif on the NetHack Learning Environment, a complex, open-ended, procedurally-generated game. The results show that agents trained with Motif's intrinsic rewards outperform those trained solely to maximize the game score. The paper also delves into the qualitative aspects of agent behavior, including alignment properties and the impact of prompt variations.

**Strengths:**

1. The paper is clear and well presented.
2. The idea of using intrinsic rewards generated from an LLM's preferences is both innovative and practically useful, potentially paving the way for more human-aligned agents.
3. The method scales well with the size of the LLM and is sensitive to prompt modifications, offering flexibility and adaptability.
4. The paper provides a comprehensive analysis, covering not just the quantitative but also the qualitative behaviors of the agents.

**Weaknesses:**

1. The paper could benefit from a more extensive comparison to other methods, especially those that also attempt to integrate LLMs into decision-making agents.
2. There is a lack of discussion on the computational cost and efficiency aspects of implementing Motif.
3. While the paper makes a strong case for Motif, it doesn't delve deeply into the limitations or potential drawbacks of relying on LLMs for intrinsic reward generation.

**Questions:**

1. Could the authors offer insights into why agents trained  on extrinsic only perform worse than those trained on intrinsic only rewards?
2. What's the best strategy to optimally balance intrinsic and extrinsic rewards during training?
3. Can the authors elaborate on the limitations of using LLMs for generating intrinsic rewards? Are there concerns about misalignment or ethical considerations?
4. How robust are agents trained with Motif against different types of adversarial attacks or when deployed in varied environments?

---

> ### Author Response · Authors · 2023-11-15
>
> Thank you for you feedback!
>
> > The paper could benefit from a more extensive comparison to other methods, especially those that also attempt to integrate LLMs into decision-making agents.
>
> First of all, we refer the reviewer Figure 7 in the Appendix F, in which we show that Motif outperforms four competitive baselines, including E3B (Henaff et al, 2022) and NovelD (Zhang et al., 2021), two state-of-the-art approaches specifically created for procedurally-generated domains such as NetHack.
>
> In the updated paper, we have now additionally added a comparison to ELLM (Du et al., 2023), a recent approach for deriving reward functions from LLMs, showing that Motif's performance significantly surpasses such LLM-based baselines across all tasks. This is due to the peculiar features of Motif's intrinsic reward (e.g., its anticipatory nature), which, by design, are not implied by a reward function based on the cosine similarity between a goal and a caption. Our implementation is described in detail in the general answer above.
>
> > There is a lack of discussion on the computational cost and efficiency aspects of implementing Motif.
>
> Due to our implementation being modular and asynchronous (that will be publicly released), dataset annotation is not especially expensive. Please see the general response for complete computational considerations. In addition, in Figure 15 we show that the performance of our method is particularly robust to the size of the dataset of annotations: Motif is able to outperform the baselines even with a dataset that is five times smaller (i.e., ~100k annotations).
>
> > While the paper makes a strong case for Motif, it doesn't delve deeply into the limitations or potential drawbacks of relying on LLMs for intrinsic reward generation.
>
> We believe, as the reviewer does, that addressing limitations in LLM-based work is critical: this is why a substantial fraction of our paper is devoted to analyzing limitations and pitfalls of intrinsic motivation from an LLM's feedback. In particular, we want to highlight that we dedicated a full page of the paper to demonstrating evidence for, explaining, and characterizing _misalignment by composition_, a negative phenomenon relevant to our framework, whose emergence is a current limitation of Motif. In addition, we studied the sensitivity of Motif to different prompts, showing in Figure 6c that semantically-equivalent prompts can lead, in complex tasks, to drastically different behaviors.  We believe this is a limitation of current approach based on an LLM's feedback, and hope that future work will be able to address it. Finally, we also included in Appendix H.3 a study on the impact of the data diversity of the dataset (through which we elicit preferences) and the resulting final performance.
>
> Please notice that we are also very upfront about the fundamental assumption behind Motif, which is also a fundamental assumption behind the zero-shot application of LLMs to new tasks: that the LLM contains prior knowledge about the environment of interest. Our Introduction is centerered around this assumption.
>
> > Could the authors offer insights into why agents trained on extrinsic only perform worse than those trained on intrinsic only rewards?
>
> Our paper already provides some insights in the "Alignment with human intuition" paragraph of Section 4.2, but we will now provide an additional perspective that can be beneficial to understand this result. By inspecting the messages preferred by the intrinsic reward function, one can quickly realize that the agent will receive from the LLM's feedback three kinds of rewards: _direct rewards_, _anticipatory rewards_ and _exploration-directed rewards_. Direct rewards (e.g., for "You kill the cave spider!") leverage the LLM's knowledge of NetHack, implying a reward very similar to the score (i.e., the extrinsic reward). Motif's reward, however, goes beyond this. Anticipatory rewards (e.g., for "You hit the the cave spider!") implicitly transport credit from the future to the past, encouraging events not rewarded by the extrinsic signal and easing credit assignment. Finally, exploration-directed rewards (e.g., for "You find a hidden door.") directly encourage explorative actions that will lead the agent to discover information in the environment. Together, these three types of rewards allow the agent to maximize a proxy for the game score that is way easier to optimize compared to the actual game score, explaining the increased performance.
>
>
> > What's the best strategy to optimally balance intrinsic and extrinsic rewards during training?
>
> We show in Figure 10c in Appendix that Motif is quite robust to how the two rewards are balanced. Broadly speaking, given that the nature of Motif's intrinsic reward brings it closer to a value function, future work can explore potentially more effective ways to leverage such type of intrinsic reward, for instance via potential-based reward shaping (Ng et al., 1999).

---

> > ### Author Response · Authors · 2023-11-15
> >
> > > Can the authors elaborate on the limitations of using LLMs for generating intrinsic rewards? Are there concerns about misalignment or ethical considerations?
> >
> > As highlighted in our previous answer to the third point, the space given to our studies on misalignment and robustness in our paper is a conscious decision. We believe this constitutes a first step in establishing this as a common practice when designing new algorithms. We want to remark here, as we did in our conclusions, that "we encourage future work on similar systems to not only aim at increasing their capabilities but to accordingly deepen this type of analysis, developing conceptual, theoretical and methodological tools to align an agent’s behavior in the presence of rewards derived from an LLM’s feedback."
> >
> > > How robust are agents trained with Motif against different types of adversarial attacks or when deployed in varied environments?
> >
> > Our experiments on prompt sensitivity (Figure 6c, 11b, 11c) can be interpreted as being close to this kind of study, showing the seemingly small variations of a prompt can trigger large or small variations of performance and behavior, depending on the environment. Future work should explore the possibility of studying the effect of actual adversarial attacks on prompts.

---

> > > ### Comment · Reviewer_VWkY · 2023-11-22
> > >
> > > Thank the authors for their clarifications and the additional experiments. Most of my concerns have been addressed. I will maintain my positive opinion towards accepting the paper.

---

### Official Review · Reviewer_wqBP · 2023-11-05

**Soundness:** 3 good
**Presentation:** 3 good
**Contribution:** 3 good
**Rating:** 8
**Confidence:** 3

**Summary:**

This paper proposes a method called Motif for tackling the exploration problem in RL by making use of an LLM to learn an intrinsic reward function, which is used to learn a policy using a standard RL algorithm. The testbed for this approach is the NetHack Learning Environment, a popular procedurally generated environment with sparse reward. This approach leads to strong improvements in performance over prior approaches without demonstrations and also results in human understandable behaviours.

**Strengths:**

- Figure 1 provides a nice clean bird's eye view of the overall approach and helps with readability.
- The evaluation of the agent for not just the game score but also other dimensions provides a helpful qualitative assessment of the proposed approach and baselines through the spider graph in Figure 4.
- The ablation experiments for the approach are quite exhaustive, covering scaling laws, prior v/s zero knowledge, rewordings of the prompts, etc.

**Weaknesses:**

- Using a 70-billion LLM to generate a preference dataset from given captions is quite expensive; while I understand this is out of the scope of the paper, perhaps using a large VLM to annotate frames without captions might have been more economical?
- Given that one of the key contributions of the paper is the intrinsic reward function that is learnt from preferences extracted from the LLM, it might be worthwhile having a baseline that gives preferences using a simpler model (say sentiment analysis) and learn the RL policy using this intrinsic reward model.

**Questions:**

- In the part on "Alignment with human intuition", the paper mentions that the agent exhibits a natural tendency to explore the environment by preferring messages that would also be intuitively preferred by humans. Is this a consequence of having a strong LLM, or is it due to the wording of the prompt?
- An ablation over $\alpha_2$ has been provided in the appendix, but the value of the coefficient for the intrinsic reward $\alpha_1$ is kept fixed at 0.1; could you explain the reason behind that?
- In Figure 6c, the score for the reworded prompt is quite low but its dungeon level keeps steadily rising compared to the default prompt. Is this a case of the agent hallucinating its dungeon level due to a very high intrinsic reward, or is it something else?

---

> ### Author Response · Authors · 2023-11-15
>
> Thank you for your feedback!
>
> > Using a 70-billion LLM to generate a preference dataset from given captions is quite expensive
>
> Due to our implementation being modular and asynchronous (that will be publicly released), dataset annotation is not especially expensive. Please see the general response for complete computational considerations. In addition, in Figure 15 we show that the performance of our method is particularly robust to the size of the dataset of annotations: Motif is able to outperform the baselines even with a dataset that is five times smaller (i.e., ~100k annotations).
>
> > while I understand this is out of the scope of the paper, perhaps using a large VLM to annotate frames without captions might have been more economical?
>
> The question of whether annotations extracted from a VLM would be more efficient than the ones extracted from running an LLM on captions is interesting. Unfortunately, our experiments with current open VLM models suggest that none of them are able yet to interpret visual frames well enough to provide an effective evaluation or even accurate captions (most likely because current open models are predominantly trained on natural images). However, given the current pace of VLM research, this may change very soon. Thus, investigating the difference in the efficiency of various types of feedback will be an interesting avenue for future research.
>
> In general, the question of large VLMs operating on images vs LLMs operating on captions brings interesting tradeoffs. Large VLM are more general, since they do not assume access to captions, but are faced with a more challenging task since they work with complex images rather than compressed text descriptions. From a purely computational standpoint, if captions are available and are of high quality then LLMs are likely more economical since their inputs are smaller.
>
> > it might be worthwhile having a baseline that gives preferences using a simpler model (say sentiment analysis) and learn the RL policy using this intrinsic reward model.
>
> We added the results of an experiment using a sentiment analysis model as a preference model in the updated paper (Figure 8 of Appendix A.6). We use a [T5 model](https://huggingface.co/mrm8488/t5-base-finetuned-imdb-sentiment) fine-tuned for sentiment analysis, and extract, for each message, a score computed as the sigmoid of the confidence of the model in its positive or negative sentiment prediction. Then, for each pair in the dataset, we assign a preference based on the message with higher sentiment score. Finally, we execute reward training and RL training as with Motif.
>
> Results on the `score` task show performance close to zero, both with and without extrinsic reward. This poor performance can be easily explained: a generic sentiment analysis model cannot capture the positivity or negativity of NetHack-specific captions. For instance, killing or hitting are generally regarded as negative statements, but they become positive in the context of killing or hitting monsters in NetHack. Llama 2 can understand this out-of-the-box without any fine-tuning, as demonstrated by our experiments. Also note that such a vanilla sentiment analysis model cannot be easily steered, thus losing any opportunity for the controllability offered by Motif.
>
> To attest to Motif's strong performance, we also compared with an additional LLM-based baseline (as requested by Reviewer VWkY). The details of this additional experiment are presented in the common response above.
>
>
> > the paper mentions that the agent exhibits a natural tendency to explore the environment by preferring messages that would also be intuitively preferred by humans. Is this a consequence of having a strong LLM, or is it due to the wording of the prompt?
>
> We believe this is due to the fact that the LLM was pretrained on massive amounts of human data, and then fine-tuned on human preferences. Indeed, even when using the zero-knowledge prompt presented in Prompt 2 of the Appendix B, Motif's reward function allows agents to play the game effectively even without any reward signal from the environment (see Figure 6b).

---

> > ### Author Response · Authors · 2023-11-15
> >
> > > An ablation over $\alpha_2$ been provided in the appendix, but the value of $\alpha_1$ the coefficient for the intrinsic reward is kept fixed at 0.1; could you explain the reason behind that?
> >
> > The important factor when combining two terms in a reward function is the relative weight given to each one of them.  We decided to ablate by varying $\alpha_2$ to progressively give more weight to the extrinsic reward, compared to the intrinsic reward. This allows us to show that, while Motif already performs well in the absence of extrinsic reward (i.e., for $\alpha_2=0$), adding progressively more importance to the reward signal coming from the environment (by increasing $\alpha_2$) correspondingly increases performance, but only up to a point. In the moment at which the relative weight given to the intrinsic reward becomes too small, the performance starts degrading as the agent acts essentially more and more as the extrinsic-only baseline.
> >
> > > In Figure 6c, the score for the reworded prompt is quite low but its dungeon level keeps steadily rising compared to the default prompt.
> >
> > Figure 6c shows that a rewording of the prompt in a task as complex as the one of finding the oracle can cause a complete change in the strategy followed by the agent. In the case of the original prompt, the agent hacks the reward, solving the task without the need of going down the dungeon. When using the reworded prompt, the agent instead starts going down the dungeon to look for the oracle, and finds it for up to 7% of the episodes. The plot thus shows the sensitivity of systems like Motif to variations of the prompts that could be perceived as small to humans. In the paper we put a strong emphasis on understanding such changes in behavior as we believe it to be fundamental if we are to release any LLM-based agent in more realistic situations. As a side note, we corrected the y axis label and normalization in Figure 6c in the updated paper to be consistent with the rest of the paper (from "Score" to "Success Rate").

---

> > > ### Author Response · Authors · 2023-11-17
> > >
> > > Dear reviewer,
> > >
> > > We believe our response addresses your concerns. In particular, we have now a total of 7 different baselines, including two additional baselines using language model-based rewards (sentiment analysis and ELLM), which are all significantly outperformed by Motif.
> > >
> > > If there are any remaining concerns which prevent you from increasing your score, please let us know in the discussion as soon as possible and we will do our best to address them.
> > >
> > > Thank you,
> > >
> > > The Authors

---

> > > > ### Comment · Reviewer_wqBP · 2023-11-22
> > > >
> > > > Thank you for the responses to my questions and also for the additional experiments. I also appreciate the overall note to all reviewers in which compute-related issues raised by all reviewers were addressed. I am satisfied with the responses and am increasing my score.

---

### Author Response · Authors · 2023-11-15

We thank the reviewers for their helpful feedback. Overall, reviewers appreciated our method, finding it innovative and useful (VWkY), interesting and very effective (2GW2), and able to lead to strong results (j8Na). They appreciated the depth of our emprical analyses and evaluations (wqBP, VWkY), as well as our presentation and visualizations (wqBP, VWkY, 2GW2, j8Na).

Reviewer wqBP and Reviewer VWkY expressed interest in understanding how computationally expensive Motif is. Running our code based on PagedAttention as implemented in vLLM (Kwon et al., 2023) on a node with eight A100s, the annotation of the full dataset of pairs takes about 4 GPU days when using Llama 2 13b and 7.5 GPU days when using Llama 2 70b. Given the asynchronous nature of our code, the required wall-clock time can be significantly reduced if additional resources are available. We have now included all this information in the paper in Appendix A.8.4. We believe that, together with the use of open models and our algorithm's robustness to data diversity and cardinality (see Appendix A.8.3), this makes experimenting with Motif affordable and accessible to many academic labs.

We will release our efficient implementation, together with the entire set of annotations used in our experiments, for the benefit of the research community. Note that, once the dataset is annotated, there is no use of the LLM anymore, and the combination of reward model training and a 1B-steps RL training run can take less than 10 hours on an A100 GPU. This also means that a policy trained with Motif can be deployed to act in real time, as long as the policy architecture runs fast enough, and regardless of the computational cost of running the LLM itself at annotation time. Our updated paper explicitly highlights these computational considerations.

Reviewer VWkY and Reviewer j8Na were interested in comparing Motif to additional approaches using LLMs for decision-making. The first  LLM-based baseline to which we compare is based on leveraging Llama 2 70b as a policy on the raw text space (similar to Wang et al, 2023). This did not lead to any performance improvement over a random baseline (we discuss this in Section F of the Appendix). The second LLM-based baseline we implemented is a version of the recently-proposed ELLM algorithm (Du et al, 2023). This implementation of ELLM closely follows the details from the paper. As intrinsic reward it uses the cosine similarity between the representation (provided by a BERT sentence encoder) of the game messages and the "early game goal" extracted from the NetHack Wiki (the first two lines from [the early game strategy](https://nethackwiki.com/wiki/Standard_strategy#The_early_game)). Despite Motif not relying on any such information from the NetHack Wiki, it significantly outperforms ELLM in all tasks. ELLM does not provide any benefit on complex tasks: its reward function cannot, by design, exhibit the exploration and credit assignment properties of the one produced by Motif (see Section 4.2 of the paper, "Alignment with human intuition"). Note that the results for the ELLM baseline are still running and we currently can only show up to 650M steps (due to its iteration speed being considerably slower than Motif). We will include the full curves in the final version of the paper.

---

### Author Response · Authors · 2023-11-21

Dear Reviewers,

we would like to thank you for taking the time to read our paper and provide reviews that have strengthened it. As the discussion phase is closing, if you have any remaining concerns we would be happy to answer them.

Best,
The Authors

---

### Meta-Review · Area_Chair_VrT2 · 2023-12-24

**Metareview:**

The paper proposes Motif, a general method to interface such prior knowledge from a Large Language Model (LLM) with an agent. Motif is based on the idea of grounding LLMs for decision-making without requiring them to interact with the environment: it elicits preferences from an LLM over pairs of captions to construct an intrinsic reward, which is then used to train agents with reinforcement learning.

###  Strengths
+ Strong experimental evaluation supports the surprising conclusion that MOTIF can outperform an agents explicitly maximizing environment reward.
+ Scaling and ablation experiments are informative, and showcase the effectiveness of LLMs for exploration.

### Weaknesses:
+ Clarity of presentation about MOTIF being intrinsic or preference based reward models.
+ Focus on NetHack as the only experimental domain
+ the presentation of the findings should be more realistic with clarity on the limitations.

**Justification For Why Not Higher Score:**

The current work received high reviews after rebuttal from the 3 reviewers, and provide a strong and insightful contribution.
Additional experimentation and a discussion on limitations are missing.

**Justification For Why Not Lower Score:**

All Reviewers are in agreement of the need for such a line of work and appreciate the framework.
Reviewers agreed on improving the scores after discussion, and the AC agrees with the consensus

---

### Decision · Program_Chairs · 2024-01-16

Accept (poster)